# Cell non-autonomy amplifies disruption of neurulation by mosaic *Vangl2* deletion in mice

Gabriel L. Galea [1,2✉], Eirini Maniou [1], Timothy J. Edwards[1], Abigail R. Marshall[1], Ioakeim Ampartzidis [1], Nicholas D. E. Greene [1] & Andrew J. Copp [1]

Post-zygotic mutations that generate tissue mosaicism are increasingly associated with severe congenital defects, including those arising from failed neural tube closure. Here we report that neural fold elevation during mouse spinal neurulation is vulnerable to deletion of the VANGL planar cell polarity protein 2 (*Vangl2*) gene in as few as 16% of neuroepithelial cells. *Vangl2*-deleted cells are typically dispersed throughout the neuroepithelium, and each non-autonomously prevents apical constriction by an average of five *Vangl2*-replete neighbours. This inhibition of apical constriction involves diminished myosin-II localisation on neighbour cell borders and shortening of basally-extending microtubule tails, which are known to facilitate apical constriction. *Vangl2*-deleted neuroepithelial cells themselves continue to apically constrict and preferentially recruit myosin-II to their apical cell cortex rather than to apical cap localisations. Such non-autonomous effects can explain how post-zygotic mutations affecting a minority of cells can cause catastrophic failure of morphogenesis leading to clinically important birth defects.

[1] Developmental Biology and Cancer, UCL GOS Institute of Child Health, London, UK. [2] Comparative Bioveterinary Sciences, Royal Veterinary College, London, UK. ✉email: g.galea@ucl.ac.uk

Neural tube (NT) defects such as spina bifida continue to affect an average of 1:1000 pregnancies globally, with higher disease burden in developing countries[1]. They arise due to failure of NT closure in the early embryo[2]. Mouse models faithfully recapitulate these conditions and have historically led to the identification of diagnostic and preventative strategies, as well as identifying genes and pathways required for NT closure[3–5]. Key genes include members of the Wnt/planar cell polarity (PCP) pathway and enzymes of folate metabolism[6], with the latter already a clinical target for primary prevention. Homozygous mutation of core PCP components reproducibly produces the most severe form of NT defect, craniorachischisis, in which much of the brain and all of the spinal cord remain open[2]. Mutations in core PCP components including *VANGL2* have been identified in human foetuses with craniorachischisis[7], and in individuals with NT defects such as spina bifida[8–10]. Nonetheless, most affected families do not receive an interpretable genetic diagnosis. One possible explanation is that disease-causing mutations may arise during post-zygotic embryonic development, rather than in the germ line, producing mosaicism that is not readily detectable with conventional genetic diagnostic methods[11,12]. Indeed, post-zygotic mutations in PCP components have recently been implicated in human spina bifida[11].

Mouse neurulation starts with narrowing and elongation of the relatively flat neural plate through PCP-dependant convergent extension. Neural folds subsequently elevate at the neural plate edges and meet at the embryonic dorsal midline, zippering to propagate closure along the length of the embryo[2]. Caudal to the initiation site, Closure 1, the region of open spinal neural folds (termed the posterior neuropore, PNP) undergoes closure through a fundamentally biomechanical process, requiring force-generating cellular behaviours such as apical constriction to elevate and appose the neural folds[13,14]. Analysis of VANGL2's roles in spinal NT closure is hindered by the fact that its homozygous deletion precludes the earlier step of closure initiation[15,16]. Heterozygous *Vangl2* mutant mice close their PNP successfully in most cases, although a very small percentage develop open spina bifida[5]. In keeping with this, heterozygous PCP gene mutations have been associated with spina bifida in humans[9,17], although the mechanisms linking such genetic findings with failed NT closure are unclear.

We recently reported that targeted deletion of *Vangl2* by *Grhl3^Cre* in a proportion of neuroepithelial cells and throughout the surface ectoderm causes distal spina bifida[14]. In that model, spina bifida was preceded by failure of neural fold elevation and biomechanical disruption of closure. The uninducible, multi-lineage recombination of *Grhl3^Cre* precluded attribution of *Vangl2*'s primary role to either the neuroepithelium or surface ectoderm. In other epithelia, PCP complexes comprise asymmetric transmembrane components between neighbouring cells, with VANGL and frizzled (FZD) located on opposing surfaces[18]. Core components are preferentially oriented on either rostral/proximal or caudal/distal cell membranes, such that tissue polarity emerges non-cell autonomously. Consequently, mislocalisation of PCP components in individual mutant cells necessarily prevents their neighbours from correctly localising the opposing complex components[19–21]. Protein-level planar-polarised orientation of VANGL2 is not apparent in the late PNP of the mouse embryo, where pseudostratified neuroepithelial cells have complex apical shapes often without predictably oriented borders[14,22]. Nonetheless, planar-polarised properties do emerge in this complex epithelium, including preferential arrangement of cell apices and supra-cellular actomyosin profiles in a medio-lateral orientation[14,22].

Actomyosin reorganisation is a well-documented outcome of VANGL/PCP signalling in the early mouse neural plate and well as *Xenopus* neuroepithelium[23–26]. Pulsed recruitment of acto-myosin to the apical surface is necessary for epithelial apical constriction in *Drosophila*[27,28]. Mechanisms of apical constriction described in different cell types require radial apical cap organisations or cortical actomyosin driving sequential junctional shrinkage[29]. Both the spatial and temporal pattern of actomyosin recruitment to epithelial apical surfaces is regulated by PCP signalling. In *Xenopus* mesoderm explants, knockdown of PCP components accelerates pulses of myosin recruitment to the apical cell cortex[30]. Shuttling of actomyosin contractility regulators towards the cell's apex is facilitated by microtubules in many different epithelia and stabilised microtubule bundles correlate with the onset of apical constriction during various morphogenetic events[31–35]. However, apically-arranged microtubules may also oppose apical constriction. Their polymerisation continuously pushes against cell junctions, curving under compressive forces from actomyosin-generated constriction[36]. Knockdown of *Vangl2* in mammalian Sertoli cells causes reorganisation of microtubules from long bundles into short, disorganised fragments[37], and in oligodendroglial cells VANGL2 increases microtubule density[38]. Thus, both the microtubule and actomyosin cytoskeletons are VANGL2 targets and contribute to apical constriction.

Neuroepithelial apical constriction has been documented in the mouse by ourselves and others[13,22,39,40]. The neuroepithelium has a predominantly apical localisation of F-actin and non-muscle myosin-II[41,42]. Pharmacological inhibition of myosin activation reduces apical neuroepithelial mechanical tension and causes PNP widening[22]. The molecular regulation of apical constriction has been extensively studied in non-mammalian vertebrates, demonstrating cell-autonomous basolateral-to-apical shuttling of myosin onto an apical F-actin network[24,30,43]. Mouse embryos have been less amenable to subcellular resolution live imaging of neurulation due to their fragility, awkward 3D shape and movement artefact partly caused by their heartbeat, which is a desirable indicator of embryo viability. Previous live imaging of early mouse embryos, prior to formation of the PNP, demonstrated that global mutation of *Vangl2* reduces neural plate apical neighbour exchange during convergent extension movements without altering basal protrusive activity[44]. We have recently optimised mouse whole-embryo static culture for live imaging of later-stage embryos to allow subcellular visualisation of zippering[45].

Here we provide mechanistic evidence that mosaicism for somatic mutations in PCP components, even with only a small minority of mutant neuroepithelial cells, is indeed sufficient to cause NT defects. We explore the hypothesis that non-cell autonomous effects amplify the morphogenetic disruption caused by mosaic mutations. We find that *Vangl2*-deleted cells prevent multiple neighbours from recruiting myosin-II to their cell cortex, thereby diminishing the apical constriction of a majority of neuroepithelial cells. Remarkably, *Vangl2*-deleted cells themselves continue to constrict apically. Nevertheless, the amplifying effect of the non-autonomous anti-morphogenetic influence of mosaic *Vangl2*-deletion leads to reduction in apical neuroepithelial mechanical tension and failure of neural fold elevation, causing spina bifida.

## Results

### Mosaic neuroepithelial *Vangl2* deletion impairs posterior neuropore closure.

We initially characterised a mouse transgenic model, in which induced deletion of *Vangl2* in a minority of neuroepithelial cells is sufficient to stall PNP closure. To our knowledge, no currently available constitutive Cre drivers selectively yet persistently lineage trace the ventral PNP

neuroepithelium. Tamoxifen-inducible CreERT2 drivers are limited because early tamoxifen administration by intraperitoneal injection prevents embryo implantation and is teratogenic[46]. We recently validated an oral tamoxifen administration protocol which robustly activates embryonic CreERT2 without impeding NT closure[47]. Using this method to induce $Sox2^{CreERT2}$ led to extensive lineage tracing of neuroepithelial cells in the closed NT, but only to a very limited extent in the open PNP given the temporally-restricted window of CreERT2 activation (Supplementary Fig. 1)[48]. In contrast, the neuromesodermal progenitor marker $Nkx1.2^{CreERT2}$ lineage traces cells in both the ventral PNP and closed NT[49] (Fig. 1a–c).

In $Nkx1.2^{CreERT2/+}$; $Vangl2^{Fl/-}$; $Rosa26^{mTmG/+}$ embryos (henceforth called Cre;Fl/−), CreERT2-induced EGFP reporter expression identifies cells in which $Vangl2$ has been deleted, 24 h after tamoxifen treatment. This produces a patchy, mosaic pattern of VANGL2 deletion in the neuroepithelium (Fig. 1d). Loss of VANGL2 protein in EGFP lineage-traced cells was confirmed by quantifying its immunolocalisation, finding that its reduction is significantly greater when one allele is pre-deleted in Cre;Fl/− embryos compared with Cre;Fl/Fl (Fig. 1e, f).

The PNP fails to close, producing pre-spina bifida lesions, in 54% (7/13) of Cre;Fl/- embryos at E10.5 compared with 0% (0/23, Fisher's exact test $p < 0.001$) of Cre;Fl/Fl embryos. Cre;Fl/Fl embryos occasionally develop dorsally flexed tails suggestive of delayed PNP closure (Supplementary Fig. 2a). In this study, Cre-negative littermate embryos were used as controls for the Cre;Fl/Fl and Cre;Fl/− genotypes. Recombined cell proportions were compared between Cre;Fl/− embryos in the dorsal, closed neural tube because cells in the ventral PNP normally translocate to this location[50] (Fig. 1g–j). $Vangl2$-deleted cells, lineage traced with EGFP, are more abundant in Cre;Fl/− embryos with pre-spina bifida lesions than those which achieved PNP closure (Fig. 1g, j). Even so, there is substantial overlap between these groups and some embryos failed to close their PNPs after losing $Vangl2$ in only 16% of neuroepithelial cells (Fig. 1g). Loss of $Vangl2$ did not change the proportion of neuroepithelial cells lineage-traced by $Nkx1.2^{CreERT2/+}$ compared with a $Vangl2$-wild-type background (Fig. 1g), indicating $Vangl2$-deleted cells were not lost from the neuroepithelium.

Pre-spina bifida lesions (persistently open PNP) reflect a pathological state, progressing to spina bifida in Cre;Fl/− embryos (Fig. 1k), making them difficult to compare to control embryos with a physiological, closed NT. Morphometric analyses were therefore performed at earlier developmental stages to identify the first quantifiable phenotype before failure of PNP closure. Dorsolateral and medial hinge points do not differ overtly between the PNPs of Cre;Fl/− and control embryos (Supplementary Fig. 2b), whereas PNP length is significantly longer in Cre;Fl/− embryos than in controls at late stages of closure (Supplementary Fig. 2c–e). Preceding this, the neural folds are less elevated in Cre;Fl/− embryos than in controls (Fig. 2a, b). Thus, the first morphometrically quantifiable tissue-level consequence of mosaic $Vangl2$ deletion is failure of neural fold elevation. Neither PNP length (Supplementary Fig. 2d) nor neural fold elevation (Fig. 2b) were significantly different between Cre;Fl/Fl embryos and Cre-negative controls, and there was no significant difference in neuroepithelial thickness between genotypes (Fig. 2c).

At the cellular level, mediolateral orientation of cell apical surfaces is a readily quantifiable planar-polarised phenotype in the mouse PNP neuroepithelium[14]. Cells in control embryos, as well as both EGFP⁺ and tdTom⁺ cells in Cre;Fl/Fl embryos had preferentially mediolaterally-oriented apical surfaces (median orientation 52−53°, Fig. 2d–f). In contrast, neither $Vangl2$-deleted (EGFP⁺) nor $Vangl2$-replete (tdTom⁺) cells showed

preferential mediolateral orientation in Cre;Fl/− embryos (median orientation 42° each, Fig. 2e). The proportions of deleted and replete cells in each orientation bracket were significantly different from control embryos (Fig. 2e). Apical orientations of $Vangl2$-deleted and $Vangl2$-replete cells did not differ significantly from each other, suggesting non-autonomous disruption of apical planar polarity. However, cell apical areas were smaller in $Vangl2$-deleted cells than in $Vangl2$-replete cells in the same Cre;Fl/− embryos (Fig. 2g). $Vangl2$ deletion did not alter neuroepithelial proliferation (Supplementary Fig. 2f, g).

**Mosaic $Vangl2$ deletion diminishes neuroepithelial apical tension.** In both mouse and chick embryos, neural fold elevation requires apical tension generated by actomyosin-dependent apical constriction[22,26], and variable apical areas in Cre;Fl/− embryos suggest differential constriction within the mosaic neuroepithelium. Cellular mechanical tension is commonly inferred from recoil of cell borders immediately following laser ablation. We previously demonstrated that annular laser ablations in the apical neuroepithelium produce actomyosin-dependant, rapid initial shrinkage of the cluster of cells within the annulus as they are untethered from the surrounding tissue[22]. Here, the reduction in apical area of cells or cell clusters following ablation is referred to as retraction to differentiate it from spontaneous apical constriction during live imaging. Apical retraction is smaller in some neuroepithelial cells in Cre;Fl/− embryos (Fig. 3a–d), but not in Cre;Fl/Fl (Supplementary Fig. 3a), compared with controls. This suggests $Vangl2$ deletion diminishes neuroepithelial apical contractility.

To corroborate this finding in an independent model, we used $Ghrl3^{Cre}$ to mosaically delete neuroepithelial $Vangl2$. $Ghrl3^{Cre}$ deletion of $Vangl2^{Fl/Fl}$ produces late depletion of neuroepithelial VANGL2 and distal spina bifida, without requiring pre-deletion of a $Vangl2$ allele[14]. Neuroepithelial expression of the $Grhl3$ gene is very low and patchy at E8.5, when it is largely restricted to the surface ectoderm overlying the neural folds, but increases at later stages[51]. Consistent with this, the knock-in $Grhl3^{Cre}$ lineage-traces a small proportion of neuroepithelial cells as early as the 8-somite stage and this proportion increases during development[14,45,52]. $Grhl3^{Cre/+}Vangl2^{Fl/Fl}$ embryos show patchy loss of neuroepithelial VANGL2 protein at later stages of development, beyond the ~19-somite stage, but not earlier (Supplementary Fig. 4a–c). Neuroepithelial annular ablations performed at earlier developmental stages, before VANGL2 protein is diminished, showed no difference in apical tension between $Grhl3^{Cre/+}Vangl2^{Fl/Fl}$ embryos and their control littermates (Supplementary Fig. 4d). In contrast, at later developmental stages when VANGL2 is predictably diminished, $Grhl3^{Cre/+}Vangl2^{Fl/Fl}$ embryos underwent less retraction following neuroepithelial annular ablation than control littermates (Supplementary Fig. 4d). Additionally, $Grhl3^{Cre}$ deletes $Vangl2$ in the surface ectoderm cells surrounding the open PNP[14]. These cells assemble long actomyosin cables which biomechanically couple the recently-fused zippering point to the caudal, open PNP[13,14] (Supplementary Fig. 4e). We previously confirmed that these cables remain present in $Grhl3^{Cre/+}Vangl2^{Fl/Fl}$ embryos[14]. In the current study, we performed cable laser ablations to test whether loss of VANGL2 diminishes contractility of these structures as well (Supplementary Fig. 4f). Cable recoil following ablation was equivalent between $Grhl3^{Cre/+}Vangl2^{Fl/Fl}$ embryos and control littermates (Supplementary Fig. 4g), suggesting that loss of VANGL2 reduces contractility selectively in the apical neuroepithelium.

**Mosaic $Vangl2$ deletion non-autonomously impairs apical constriction.** In Cre;Fl/− embryos, the reduction in apical

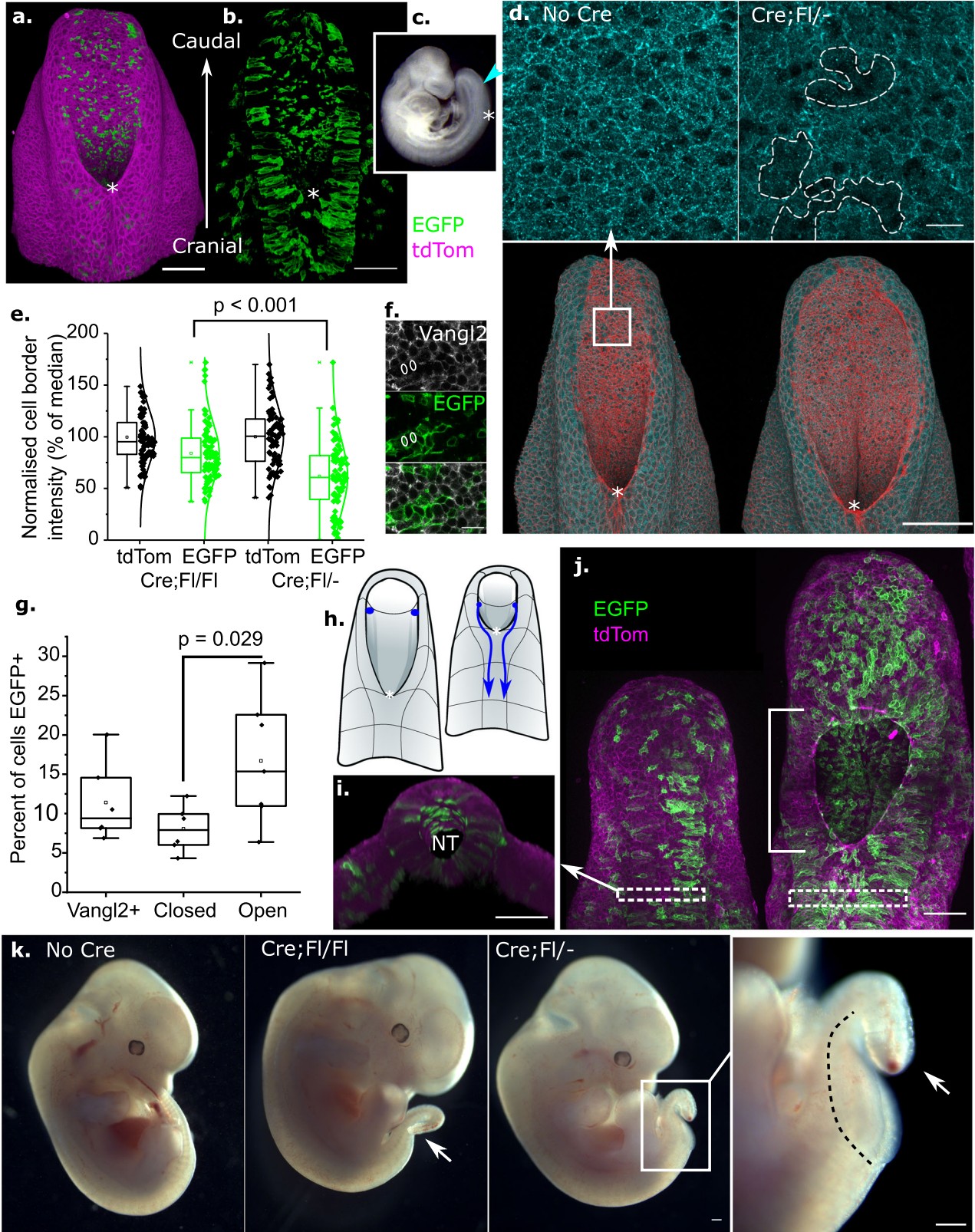

retraction following annular ablation is limited to cell clusters which include *Vangl2*-deleted cells. Selective annular ablations which do not include *Vangl2*-deleted cells produce equivalent retractions to control embryos (Fig. 3c). This shows apical tension is diminished locally in association with the *Vangl2*-deleted cells. To test whether the local diminution of neuroepithelial apical retraction around *Vangl2*-deleted cells occurs cell autonomously,

apical areas of *Vangl2*-deleted EGFP⁺ cells were analysed separately from their *Vangl2*-replete tdTom⁺ neighbours. Whereas *Vangl2*-deleted cells were found to retract similarly to cells in control embryos, their *Vangl2*-replete neighbours retracted less (Fig. 3e–g). Retraction magnitude was not correlated with the number of EGFP⁺ cells in each cluster (Supplementary Fig. 3b, c). Both EGFP⁺ cells and tdTom⁺ neighbours retracted similarly to

**Fig. 1 Mosaic *Vangl2* deletion causes spina bifida. a, b** PNP 3D reconstructions illustrating Nkx1.2$^{CreERT2}$ lineage tracing using the mTmG reporter. In **b** non-recombined cells which express tdTom are not shown. Asterisk (*) indicates the zippering point. **c** Bright field image of E9.5 mouse embryo indicating the zippering point (*) and PNP (arrowhead). **d** Surface-subtracted wholemount immunolocalisation of VANGL2 in the apical neuroepithelium of control (No Cre) and Cre;Fl/- embryo, showing mosaic deletion (white dotted lines) in the latter. Scale = 20 μm. Whole-PNP images are 3D reconstructions. **e** Quantification of VANGL2 immunolocalisation on cell borders lineage traced with EGFP or tdTom in Cre;Fl/Fl and Cre;Fl/− embryos. Average VANGL2 intensity in non-recombined (tdTom) cells was set at 100% in each embryo (four embryos from independent litters per genotype). *p* values from ANOVA with post-hoc Bonferroni. **f** Optical cross-section through the neuroepithelium of a Cre;Fl/− embryo showing endogenous EGFP and immunofluorescently-detected VANGL2. White circles indicate borders between two EGFP+ cells, which are devoid of Vangl2 signal. Scale = 20 μm. **g** Quantification of the proportion of recombined neuroepithelial cells in E10.5 Cre;Fl/− embryos which closed (*n* = 6) or failed to close (*n* = 7) their PNP. VANGL2+ indicates the proportion of recombined cells in *Nkx1.2$^{CreERT2/+}$Vangl2$^{+/+}$Rosa26$^{mTmG/+}$* embryos with wild-type VANGL2 (*n* = 6). *p* value from Students *T*-test. **h** Early (left) and late (right) PNPs illustrating progressive shortening and narrowing during physiological closure. Cells in the caudal, open PNP (blue dots) translocate to the dorsal, closed PNP (blue arrows) during closure. **i** 3D cross-section of the dorsal closed NT illustrating the region where EGFP+ neuroepithelial cells were analysed. **j** Images of two E10.5 Cre;Fl/− embryos with low (left) and high (right) recombination assessed from EGFP expression. Left: 37 somites, successful PNP closure. Right: 39 somites, tile-scan of a pre-spina bifida lesion (white bracket). Dashed white boxes denote the closed neural tube where recombination was quantified. **k** Phenotypes of Cre;Fl/Fl and Cre;Fl/− embryos at E12.5. Cre;Fl/Fl embryos developed dorsally kinked tails (arrow). Cre;Fl/− embryos also develop spina bifida (dashed line, seen in embryos from three independent litters). Scale = 100 μm unless otherwise indicated.

controls in Cre;Fl/Fl embryos (Supplementary Fig. 3d). Thus, the diminution of apical retraction following mosaic *Vangl2* deletion is non-cell autonomous: *Vangl2*-replete neighbours of *Vangl2*-deleted cells fail to undergo apical retraction, whereas the *Vangl2*-deleted cells themselves continue to retract despite most of them neighbouring other EGFP+ cells.

To expand this analysis of neighbouring cells, we analysed the apical area of *Vangl2*-replete cells neighbouring different numbers of *Vangl2*-deleted cells using a semi-automated segmentation pipeline in fixed embryos (Supplementary Fig. 5a). This confirmed that *Vangl2*-deleted cells have smaller apical areas than their neighbours, but neighbours' apical areas were not correlated with the number of *Vangl2*-deleted cells they touched (Supplementary Fig. 5b, c). Thus, we do not find evidence of a dose-dependent influence of the number of *Vangl2*-deleted cells contacted on the apical properties of *Vangl2*-replete neighbours. Other morphometric comparators of apical shape quantified were not significantly different between cell groups (Supplementary Fig. 5d–f).

Potential explanations for the observed differences in apical area and retraction following laser ablation include changes in cell adhesion, material properties, or actomyosin-dependent constriction. *Vangl2*-deleted cells and their neighbours continue to assemble adherens junctions labelled with N-cadherin (CDH2) and active β-catenin (CTNNB1), as well as tight junctions labelled with ZO1 and an apical F-actin cortex (Supplementary Fig. 6a–e), suggesting that differences in retraction are not caused by differential cell adhesion.

We next sought to directly assess apical constriction through visualisation in live-imaged embryos. The length of live-imaged sequences was limited by substantial changes in tissue morphology as the PNP continued to narrow (Supplementary Fig. 7a) and time constraints of imaging at least one control and one comparable Cre;Fl/- embryo from each litter, while avoiding prolonged culture. Live-imaged sequences were therefore limited to 20 min. Over this time, neuroepithelial cells were found to vary their apical surfaces in an asynchronous, frequently pulsatile manner characteristic of apical constriction in other cell types (Supplementary Fig. 7b–d). Individual cells could be in constriction or dilation phases at the start of imaging. In order to rationalise the data, the apical sizes of individual cells were temporally aligned by their largest observed apical area. This produced averaged traces of dilation followed by constriction (Supplementary Fig. 7b–d). A pilot study analysing a wild-type embryo showed that 24 cells need to be analysed to detect a 20% difference in apical area reduction (*p* = 0.05, power = 0.8).

*Vangl2*-deleted cells in Cre;Fl/− embryos dilate faster than cells in control embryos, but subsequently constrict at a similar rate (Fig. 4a–c). This pattern could be explained by either a change in constriction frequency or magnitude, but we cannot discriminate between these with the available data (Supplementary Fig. 7e). In contrast, *Vangl2*-replete neighbours of *Vangl2*-deleted cells dilate similarly to controls, but then fail to constrict (Fig. 4d). Changes in apical areas of *Vangl2*-deleted and neighbouring cell pairs were not predictably correlated, showing that *Vangl2*-replete neighbours do not dilate as *Vangl2*-deleted cells constrict (Supplementary Fig. 8). *Vangl2*-replete cells which do not contact *Vangl2*-deleted cells in Cre;Fl/− embryos (Distant cells in Fig. 4d) dilate and constrict similarly to cells in control embryos. These findings corroborate the laser ablation analyses by implicating non-autonomous failure of apical constriction in *Vangl2*-replete neighbour cells, which leads to diminution of overall neuroepithelial apical tension.

Three subgroups of cells were defined for further analysis: 1) *Vangl2$^-$*/EGFP+ cells which constrict, 2) *Vangl2$^+$*/EGFP$^-$ neighbouring cells which do not constrict and, 3) *Vangl2$^+$*/EGFP$^-$ distant cells which do constrict (Fig. 4e).

**Mosaic *Vangl2* deletion alters the actomyosin and microtubule cytoskeletons.** Cytoskeletal regulation by VANGL/PCP signalling is well established in other contexts, and both actomyosin and microtubule changes may underlie differential apical constriction in the neuroepithelium following mosaic *Vangl2* deletion[23–25,37,53,54]. Neuroepithelial cells assemble apical phosphorylated, active non-muscle myosin light chain-II around their cell cortex (Fig. 5a). In addition, phospho-myosin light chain (pMLC)-II decorates the apical cap of individual neuroepithelial cells (Fig. 5a). This pattern is accentuated when total myosin heavy chain (MHC)-IIb is visualised, producing marked differences in localisation to the cell cortex, with or without staining on apical caps (Fig. 5b). Mosaic *Vangl2* deletion did not significantly change overall MHC-IIb staining intensity (Supplementary Fig. 9a, b), but significantly reduced the variability of apical cap staining across the neuroepithelium (Supplementary Fig. 9c).

The effect of mosaic *Vangl2* deletion on both cortical and apical cap MHC-IIb was therefore assessed sequentially. The same distinct cortical and apical cap staining pattern is observed with a second commercial anti-MHC-IIb antibody (Supplementary Fig. 10a). MHC-IIb staining visualised on cell borders is the average between adjacent cells: 1) EGFP/neighbour borders are between a contractile and a non-contractile cell, 2) neighbour/neighbour borders are

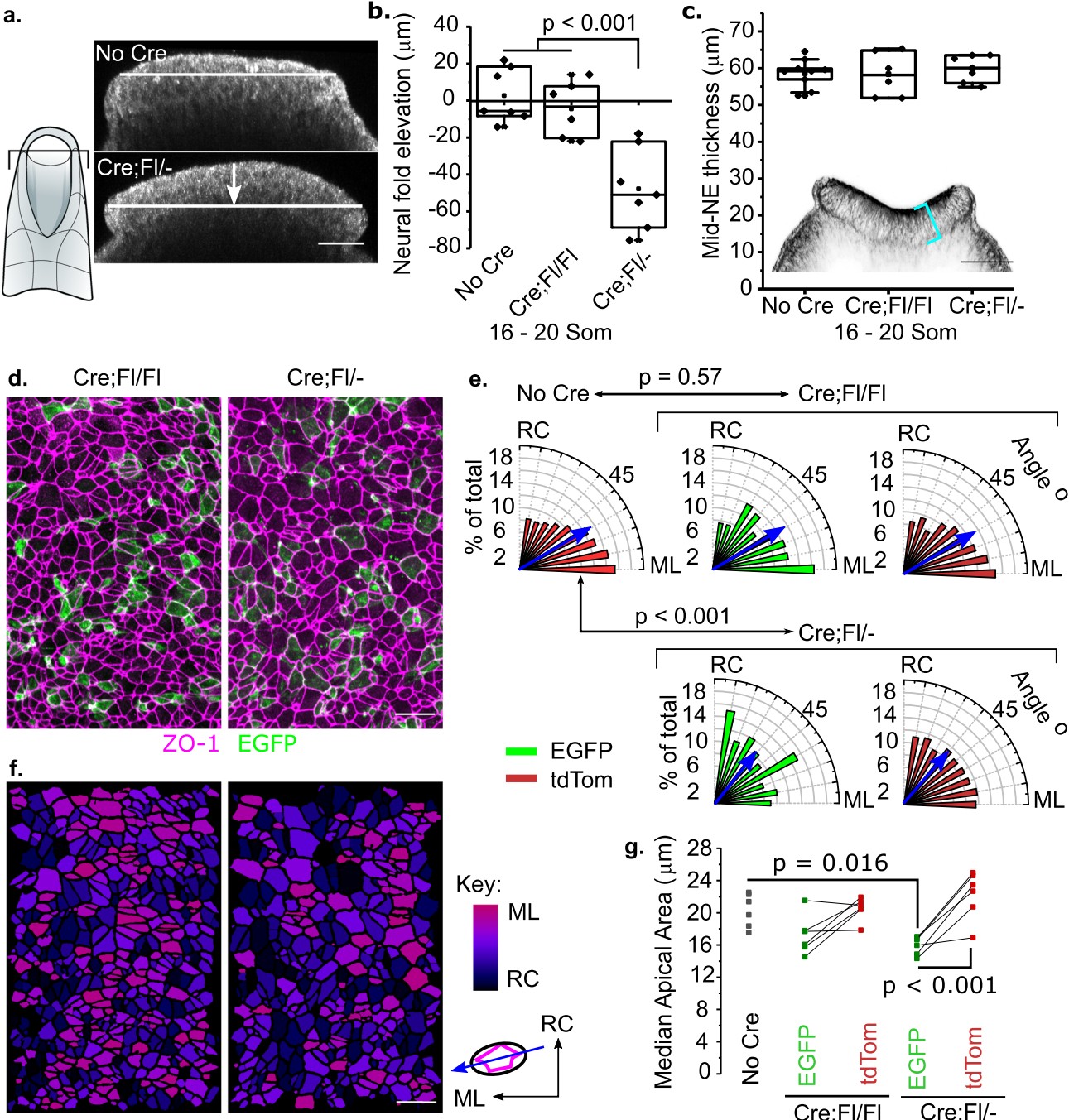

**Fig. 2 Mosaic *Vangl2* deletion disrupts planar polarised neuroepithelial apical orientation. a** Optical cross-sections through the caudal PNP (at 75% of rostrocaudal PNP length, shown in schematic) of No Cre and Cre;Fl/− embryos at E9.5, visualised using reflection imaging. The white arrow indicates the quantified eversion of the apical neuroepithelium, which should elevate its lateral margins to form neural folds. Scale = 50 μm. **b**, **c** Quantification of neural fold elevation (**b**, positive values) or eversion (**b**, negative values) and mid-PNP neuroepithelial thickness (**c**) in control, Cre;Fl/Fl and Cre;Fl/− embryos at 16–20 somite stages, before the PNP is significantly longer in Cre;Fl/− embryos. The inset shows an optical cross-section through a phalloidin-stained PNP indicating the thickness of the neuroepithelium (cyan bracket). **b** Control = 7, Cre;Fl/Fl = 6, Cre;Fl/− = 7 embryos. **c** Control = 9, Cre;Fl/Fl = 6, Cre;Fl/− = 7 embryos. Scale = 50 μm. **d** Representative (>6 embryos) surface-subtracted images of the apical neuroepithelium in a Cre;Fl/Fl embryo and Cre;Fl/− littermate showing the apical marker ZO1 and recombined cells lineage traced with EGFP. Scale = 20 μm. **e** Quantification of the orientation of the apical long axis of neuroepithelial cells in No Cre, Cre;Fl/Fl and Cre;Fl/− embryos. Blue arrows indicate the median orientation; *p* values from Chi[2] with Yates continuity correction; EGFP and tdTom not significantly different within genotypes. No Cre 5592 cells from six embryos; Cre;Fl/Fl EGFP/tdTom 318/2373 cells from five embryos; Cre;Fl/− EGFP/tdTom 267/3799 cells from six embryos. RC rostrocaudal, ML mediolateral. **f** Apical surfaces from **d** colour coded to illustrate orientation. Bright colours indicate medio-lateral, dark colours indicate rostro-caudal orientation. Scale = 50 μm. **g** Quantification of neuroepithelial apical area. Each point represents the median value for an embryo and lines link points from the same embryo, *n* = 6 embryos per genotype. *p* values from ANOVA with post-hoc Bonferroni. Embryos compared had 15–20 somites and had been administered tamoxifen 24–28 h prior to analysis.

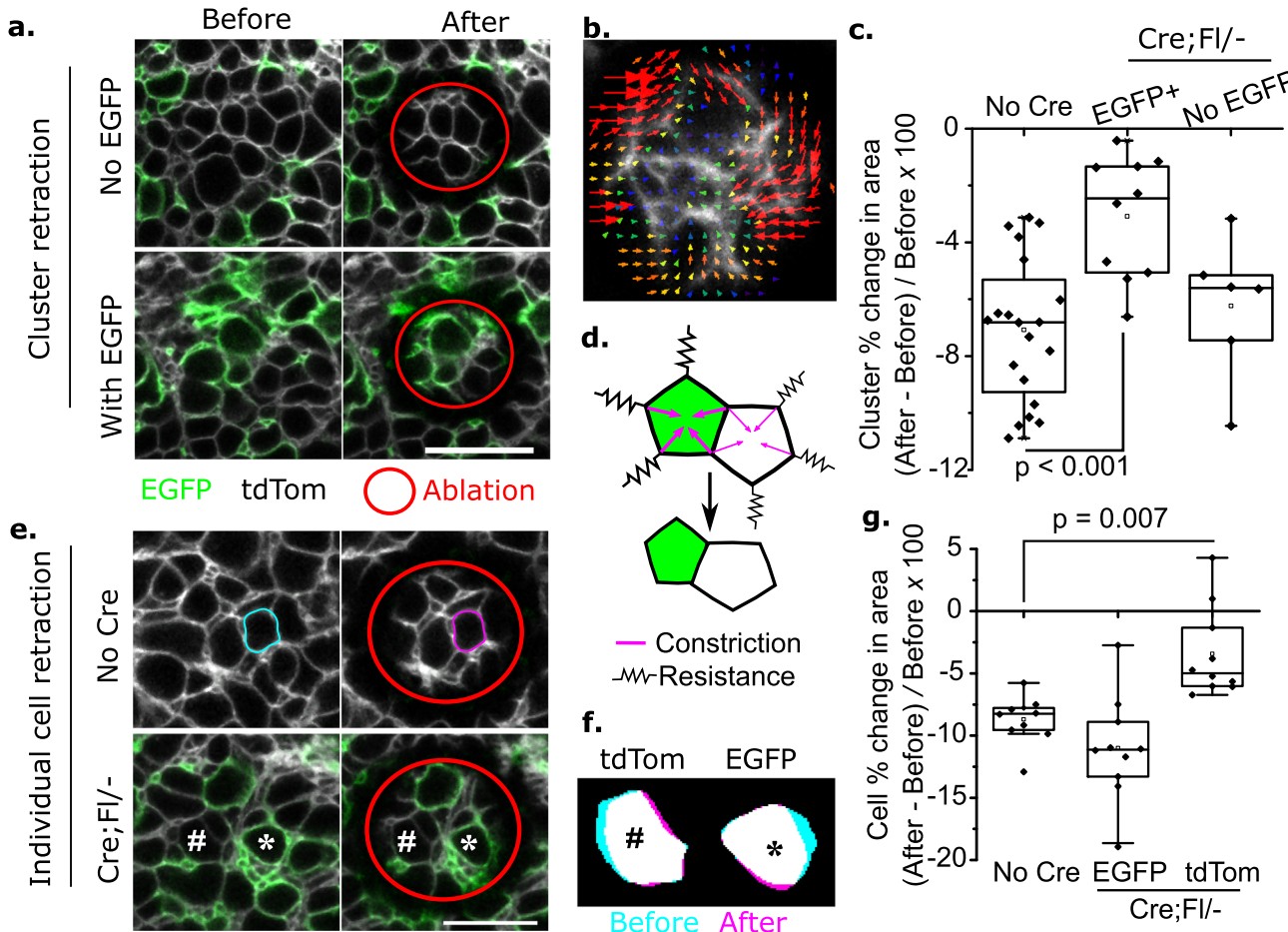

**Fig. 3 Mosaic *Vangl2* deletion non-autonomously diminishes neuroepithelial apical retraction. a** Representative (>6 embryos) live-imaged neuroepithelium before and after annular laser ablation. Ablations were made in regions including (with EGFP) or excluding (no EGFP) *Vangl2*-deleted cells. **b** Particle image velocimetry (arrows superimposed on grey cell borders) illustrating apical retraction of a cluster of cells within the ablated annulus. **c** Quantification of the retraction (% change in area) of cell clusters within the ablated annulus. Dots represent a cluster in an individual embryo; each embryo was only ablated once. Control $n = 20$, EGFP+ $n = 10$, No EGFP $n = 6$ embryos. **d** Schematic illustration of cell retraction due to immediate constriction when released from connections to surrounding cells. **e, f** Representative live-imaged neuroepithelium from control (No Cre, 20 embryos) and Cre;Fl/− (10 embryos). A cell is traced before and after ablation in the No Cre control. The # and * correspond to the cells shown in **f**, illustrating retraction. **g** Quantification of the retraction (% change in area) of individual cells within the ablated annulus. Ablations included 2–3 *Vangl2*-deleted (EGFP) and *Vangl2*-replete (tdTom) neighbours of the deleted cells. Retractions of cells of the same type were averaged in each cluster, $n = 10$ embryos per genotype. Scale bars = 20 μm. *p* values from ANOVA with post-hoc Bonferroni.

between two non-contractile cells, and 3) distant/distant borders are between two contractile cells. Average MHC-IIb staining intensity was found to be highest along the most contractile distant/distant borders of Cre;Fl/- embryos, whereas EGFP/neighbour and neighbour/neighbour borders had significantly lower intensity that did not differ between them (Fig. 5c).

MHC-IIb is enriched on the apical cap of approximately half of PNP neuroepithelial cells in wild-type embryos (Fig. 5d–e). Apical cap MHC-IIb forms a pattern of punctate staining or linear arrangements resembling stress fibres (Fig. 5d cap). However, *Vangl2*-deleted cells show primarily cortical, rather than apical cap MHC-IIb staining (Fig. 5e). This effect is cell-autonomous as neither neighbouring nor distant *Vangl2*+/EGFP− cells in Cre; Fl/− embryos are different from controls (Fig. 5e).

Apical cap MHC-IIb arrangements have been described in insect cells, and are enhanced in mammalian cells with diminished microtubule turnover[55]. *Vangl2* deletion alters microtubule organisation in other mammalian cells[37,38]. Non-mitotic PNP neuroepithelial cells were found to assemble microtubules in two predominant patterns: apically-enriched radiating fibres versus apicobasally-elongated tails (Fig. 5f). Both apical pools and the elongated tails stain positively for the stable microtubule marker acetylated α-tubulin (Supplementary Fig. 10b). Both of these microtubule arrangements influence apical constriction in other contexts: apical networks counteract, whereas apicobasal tails promote constriction[34–36,56].

Apical radiating microtubule fibres were mainly seen in neuroepithelial cells with apical cap myosin, whereas elongated tails were found to be associated with a primarily cortical myosin distribution (Fig. 5f). The small proportion of neuroepithelial cells in mitosis at any one time have a tubulin network which is apical to and distinct from the centrosomal network (Supplementary Fig. 10c). Both microtubule patterns are preserved in neuroepithelia with mosaic *Vangl2* deletion, but the length of microtubule tails was found to be shorter in Cre;Fl/− embryos than in wild-type littermates (Fig. 5g). This effect appears non-autonomous as *Vangl2*-deleted cells have longer tails than their neighbours (Fig. 5h, i). Conversely, *Vangl2*-deleted cells have less abundant apical microtubule networks than their neighbours (Fig. 5j, k).

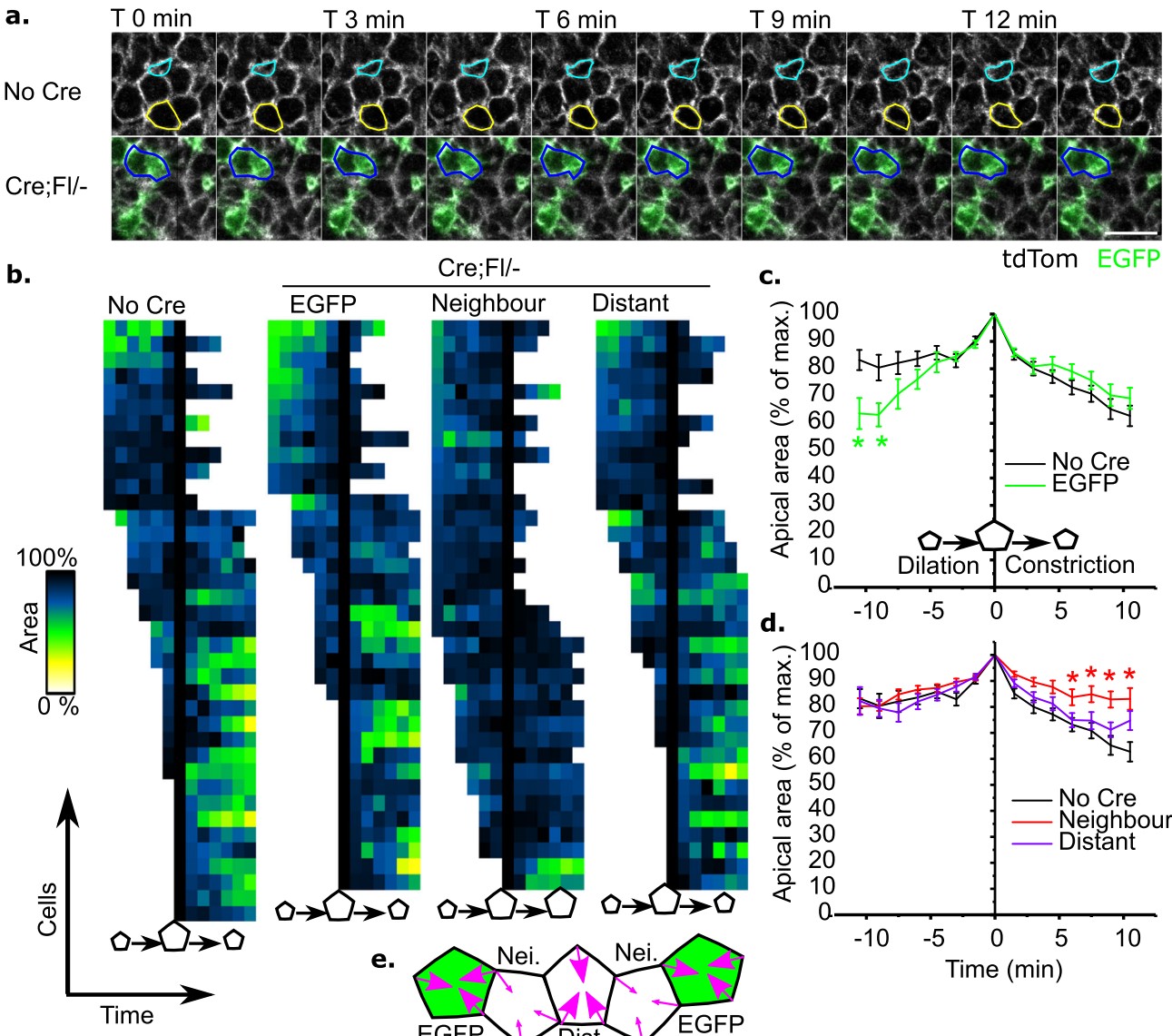

**Fig. 4 Vangl2-replete neighbours of Vangl2-deleted cells fail to apically constrict. a** Representative live-imaged sequences of No Cre (n = 6) and Cre;Fl/− (n = 6) embryos. Cyan: apically-expanding cell. Yellow: apically constricting cell. Blue: constricting then partly dilating Vangl2-deleted (EGFP+) cell. Scale bar = 20 μm. **b** Heat maps illustrating the change in apical areas of each analysed cell in No Cre and Cre;Fl/− embryos. Dark colours indicate large apical areas (up to 100% of observed sizes). **c, d** Quantification of changes in cells' apical areas during live imaging. **c** No Cre compared with EGFP recombined cells. **d** No Cre compared with cells neighbouring EGFP recombined cells, and more distant cells that do not contact EGFP recombined cells. Each cell is aligned at T = 0 when it reaches the biggest observed apical area. Negative time points represent increasing apical areas (dilation), positive time points represent constriction. Asterisk (*) indicate p < 0.05 at each indicated time point by mixed model analysis with Bonferroni post-hoc. No Cre: 38 cells from five embryos, Cre;Fl/− 36 cells of each type from five embryos. Points represent the mean ± SEM. **e** Schematic illustration of the live imaging and laser ablation results. EGFP+Vangl2− cells constrict/retract similarly to cells in control embryos. EGFP−Vangl2+ neighbours (Nei.) of Vangl2-deleted cells do not constrict/retract. EGFP−Vangl2+ distant (Dist.) cells which do not contact Vangl2-deleted cells constrict/retract normally. Magenta arrow thickness indicates constriction magnitude.

Thus, Vangl2-deleted cells preferentially localise myosin to the contractile cell cortex in a cell-autonomous manner, while both autonomously (less dense apical microtubules) and non-autonomously (shorter apicobasal tails) altering microtubule organisation in a pattern expected to favour their constriction (Fig. 5l). One mechanism by which VANGL2 may regulate both the actomyosin and microtubule cytoskeleton is through Rho-associated protein kinase (ROCK). We previously reported that the commonly-used compound Y27632 diminishes Rho/ROCK signalling by reducing active Rho in mouse whole embryo culture, preventing down-stream responses including restricted apical localisation of actomyosin and apical constriction of the neuroepithelium[22,41]. Our previously-published time course experiments show that Y27632 treatment begins to cause PNP widening within two hours of treatment in mouse whole embryo culture[22]. In the current study, we observed that 2 h of ROCK inhibition with 10 μM Y27632 markedly increases apical microtubule staining while shortening apicobasal tails of PNP neuroepithelial cells in wild-type embryos (Fig. 6a–d). ROCK1 is primarily localised around cell borders and multicell junctions, with low levels variably decorating the apical cap (Fig. 6e). Its staining intensity is significantly lower in the apical neuroepithelium of Cre;Fl/- embryos than their littermate controls (Fig. 6e, f). ROCK1 cortical staining intensity is not significantly different between the cell types compared (Fig. 6g), but

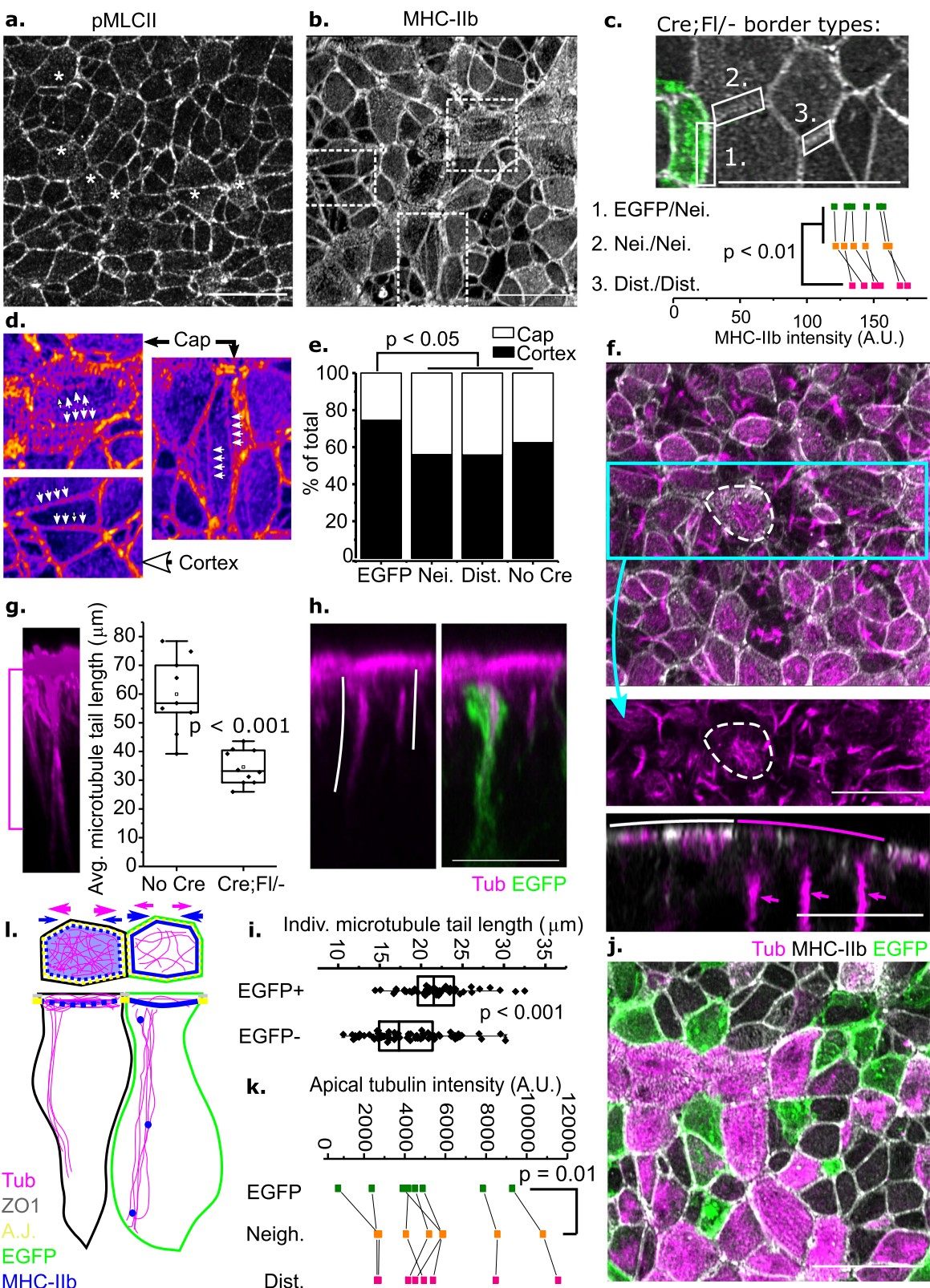

**a.** pMLCII

**b.** MHC-IIb

**c.** Cre;Fl/- border types:

1. EGFP/Nei.
2. Nei./Nei.
3. Dist./Dist.

p < 0.01

MHC-IIb intensity (A.U.)

**d.** Cap — Cortex

**e.** p < 0.05 | Cap / Cortex
% of total
EGFP Nei. Dist. No Cre

**f.**

Tub MHC-IIb EGFP

**g.** Avg. microtubule tail length (μm)
p < 0.001
No Cre    Cre;Fl/-

**h.** Tub EGFP

**i.** Indiv. microtubule tail length (μm)
EGFP+
EGFP-
p < 0.001

**j.**

**k.** Apical tubulin intensity (A.U.)
EGFP
Neigh.
Dist.
p = 0.01

**l.** Tub / ZO1 / A.J. / EGFP / MHC-IIb

*Vangl2*-deleted cells have more intense apical cap staining than either their neighbours or distant *Vangl2*-replete cells (Fig. 6h, i). Thus, VANGL/PCP signalling may influence the microtubule cytoskeleton through ROCK to promote apical constriction (Fig. 6j), in addition to ROCK's well-established regulation of neuroepithelial apical actomyosin localisation[22,41].

**Non-autonomy amplifies mosaic *Vangl2* deletion**. Having demonstrated non-autonomous suppression of apical constriction by *Vangl2*-deleted cells, we sought to quantify the potential for effect amplification as a result of each deleted cell inhibiting multiple neighbours. Lewis' law predicts that the apical area of epithelial cells should correlate linearly with the number of cells

**Fig. 5 Autonomous and non-autonomous cytoskeletal changes may underlie diminished apical constriction. a, b** Surface-subtracted neuroepithelial pMLCII (**a**) and total MHC-IIb (**b**) wholemount images. * apical staining. Dashed boxes indicate the cells shown in **d**. **c** Quantification of MHC-IIb intensity along 1: EGFP-Neighbour, 2: Neighbour–Neighbour, 3: Distant–Distant. Lines indicate values from the same embryo ($n = 6$ per genotype), compared by repeated measures ANOVA. **d** Representative cells showing apical cap arrangements (Cap) or primarily cortical (Cortex) MHC-IIb staining seen in >20 embryos. Fire LUT. Arrows indicate two types of MHC-IIb staining. **e** Quantification of the proportion of cells with primarily cortical or apical cap MHC-IIb. $p$ values by $X^2$. Cre;Fl/− EGFP = 98 cells, neighbour = 175, distant = 156 cells from six embryos; No Cre = 711 cells from six embryos. **f** Maximum-projection showing neuroepithelial MHC-IIb and β-tubulin (Tub) staining. Tubulin in the cyan boxed region is shown below. Cells with apical cap MHC-IIb have apical microtubules (cell outlined by dashed line). Bottom panel: optical cross-section showing apical cap myosin (white arc) overlies cells without long tubulin tails (magenta arrows/arc). **g** Quantification of average tubulin tail length in No Cre ($n = 9$) and Cre;Fl/− ($n = 10$) embryos. Points represent individual embryos. Magenta bracket indicates the length of tubulin tails. $p$ value from Student's $T$-test. **h** Maximum-projected optical cross-section showing tubulin tails in a Cre; Fl/− embryo. White lines indicate tail length. **i** Tubulin tail lengths in *Vangl2*-deleted (EGFP+) and *Vangl2*-replete (EGFP−) neuroepithelial cells. Points represent individual tails (EGFP = 64 tails, tdTom = 72 tails from four embryos from four independent litters). $p$ value from Mann–Whitney $U$-test. **j** Representative (six embryos) surface-subtracted apical neuroepithelium of a Cre;Fl/− embryo showing apical tubulin, MHC-IIb and EGFP staining. **k** Tubulin staining intensity on the apical cap of surface-subtracted *Vangl2*-deleted (EGFP+), neighbouring and distant (EGFP−) cells. Lines indicate intensities from the same embryo ($n = 8$), compared by repeated measures ANOVA. **l** Schematic of cytoskeletal features in *Vangl2*-deleted (green) and neighbouring *Vangl2*-replete cells. Arrows = pro-constriction (blue) and opposing force (magenta). Images obtained using AiryScan; scale bars = 20 μm.

they neighbour, and that the average cell should have six apical neighbours[57,58]. In agreement with this law, *Vangl2*-deleted cells have 5.7 total neighbours on average, of which 5.3 are *Vangl2*-replete (Fig. 7a, b). The proportion of *Vangl2*-replete cells neighbouring a *Vangl2*-deleted cell is greater in embryos with more recombined cells within the range of recombination achievable with *Nkx1.2^CreERT2* (Fig. 7c). However, cells share neighbours, so the number of unique neighbours per *Vangl2*-deleted cell decreases as the proportion of deleted cells increases (neighbours/deleted cell shown in Fig. 7d). In other words, as more cells lose *Vangl2* they each have fewer unique neighbours. Thus, neighbour-sharing limits amplification of effect size through non-autonomy and individual *Vangl2*-deleted cells have the greatest impact in embryos with low levels of recombination.

To test this, we investigated correlations between the proportion of *Vangl2*-deleted cells versus neural fold eversion, the earliest phenotype detected in Cre;Fl/− embryos. The proportion of *Vangl2*-deleted neuroepithelial cells does not correlate with magnitude of neural fold eversion (Fig. 7e). However, the proportion of *Vangl2*+/EGFP− neighbour cells is significantly correlated with neural fold eversion (Fig. 7f). This supports a model in which non-autonomous inhibition of neighbour cells' apical constriction underlies failure of PNP closure following mosaic *Vangl2* deletion.

## Discussion

Post-zygotic somatic mutations are pervasive[59,60]. Human foetal forebrain neural progenitors accumulate 8.6 new genomic variants per cell division and each harbours up to 12 non-benign variants[61]. The importance of somatic mosaicism is clinically established in dermatology, causing both malignant transformation and benign phenotypes[62]. One such example is epidermal naevus syndrome, which affects 1–3:1000 births[63]. This prevalence is comparable to that of NT defects. Post-zygotic mutations have been implicated in various congenital malformations[64,65] and mutations in PCP components have specifically been identified in individuals with NT defects[11]. Here we demonstrate, in a mammal, that these mutations only need to disrupt PCP in a small percentage of neuroepithelial cells, around 16%, to cause spina bifida.

Presumptive spinal cord neuroepithelial cells are thought to primarily derive from neuromesodermal progenitors which persist throughout neurulation[66,67]. Thus, de novo mutations in this population would be propagated to their daughter cells throughout PNP closure. We took advantage of this property in the current study to persistently delete *Vangl2* in a variable proportion of neuroepithelial cells using *Nkx1.2^CreERT2*. Resulting

deletion of *Vangl2* occurs relatively late in neurulation, after the convergent extension movements which initiate NT closure at E8-8.5. This Cre-driver does not itself impair NT closure even when homozygous[49], nor when used to delete *Rac1*[52]. In the current study, a single administration of tamoxifen only predictably deleted *Vangl2* when one allele was pre-deleted. Single administration was critical to ensure predictable deletion dynamics such that EGFP lineage tracing could be used to identify *Vangl2*-deleted cells. Temporally restricted CreERT2 activation by tamoxifen is also shown here by the inability of *Sox2^CreERT2* to persistently lineage trace the ventral PNP, despite strong expression of its endogenous gene in this tissue[48]. The lack of temporal control over *Grhl3^Cre* activation precludes its use to identify *Vangl2*-deleted cells, but this Cre driver recapitulates phenotypes observed with *Nkx1.2^CreERT2* without requiring pre-deletion of a *Vangl2* allele.

Pre-deletion of a *Vangl2* allele followed by CreERT2-mediated recombination represents two genetic hits. It is less likely that two somatic mutations would occur in the same gene, in the same cell, spontaneously. However, double heterozygous mutations of different PCP components is sufficient to cause NT defects[68] and digenic variants in PCP components are readily documented in human patients[17]. Somatic mutations in other PCP membrane complex components, including *FZD6* and *CELSR1*, have also been identified in individuals who have spina bifida[11], suggesting that the effects documented here may extend beyond *VANGL2*.

Human mutations in core PCP genes disrupt this pathway when engineered into *Drosophila*[69]. Mammalian embryonic tissues achieve planar polarity through ligand gradient-sensing, as in the case of Fat/Dachsous PCP, or through non-autonomous propagation of directionality through asymmetrical complex organisation characteristic of the 'core' PCP pathway studied here[18,70]. In *Drosophila* epithelia, asymmetrical PCP localisation emerges gradually from seemingly-homogeneous initial distributions[71,72]. This may also happen in mammals, for example as indicated by the finding of more apparent *Vangl2* planar polarity at later stages of oviduct maturation in mice[73]. Symmetry-breaking of membrane complexes may be induced by mutual repulsion between core components, directional transendocytosis of *Vangl2* between adjacent cells, or extrinsic cues such as mechanical tension[18,70,71]. Immunolocalisation of mammalian PCP components is notoriously limited by the unavailability of adequate antibodies, although the knockout-validated 2G4 anti-VANGL2 antibody[74] used here allows robust detection of VANGL2 in the mouse PNP. *Vangl1* is not widely expressed in the mouse PNP neuroepithelium[75]. This makes *Vangl2* a

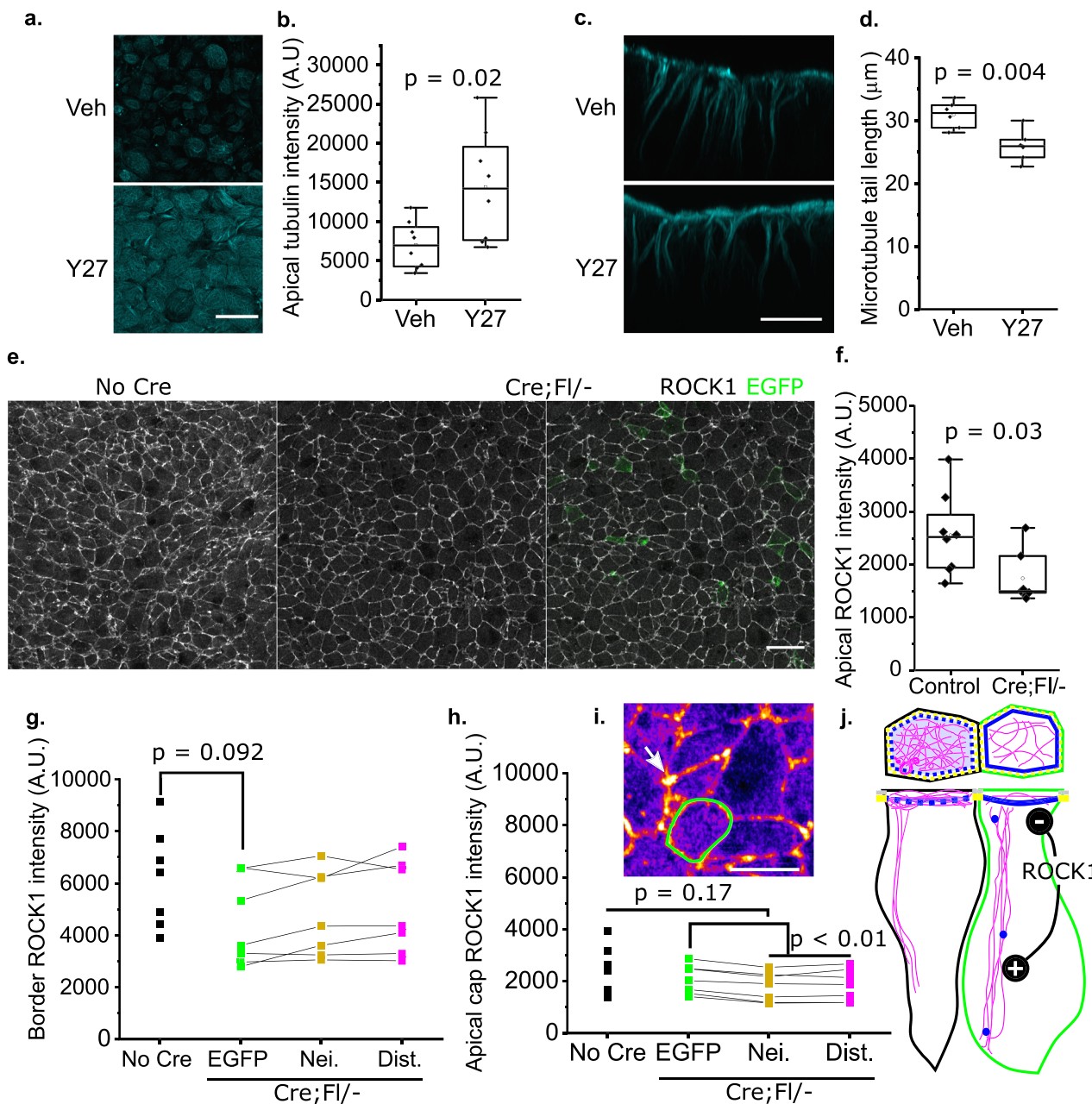

particularly valuable target to explore the functions of PCP signalling during PNP closure. Ubiquitous *Vangl2* deletion causes failure of convergent extension migration at the onset of neurulation[15] whereas, after the PNP forms, neuroepithelial cells migrate laterally and dorsally to form the dorsolateral hinge points[50]. *Vangl2* appears to control convergent extension cell-autonomously. In chimeric embryos containing *Vangl2*-mutant cells, the wild-type cells self-segregate during early convergent extension of the neural plate[15]. This contrasts with non-autonomous control of apical constriction described here. *Vangl2* deletion also did not significantly change the proportion of cells lineage traced by *Nkx1.2^{CreERT2}* in the dorsal closed neural tube, which suggests that this translocation event is regulated differently from convergent extension.

While the role of PCP signalling in polarised junction shrinkage which underlies convergent extension is well established, its role in regulating apical constriction is incompletely understood. Substantial progress has been made by studying chick neurulation, where knockdown of PCP core components results in larger neuroepithelial apical sizes and failure of neural plate dorsal bending consistent with failed apical constriction[26], as seen in mice in this study. Neuroepithelial apical tension is largely unchanged in *Nkx1.2^{CreERT2/+}Vangl2^{Fl/Fl}* embryos despite mild yet significant reduction in VANGL2 immunolocalisation. Similarly, *Grhl3^{Cre/+}Vangl2^{Fl/Fl}* embryos retain neuroepithelial apical tension at earlier developmental stages when this non-inducible Cre abundantly lineage-traces the neuroepithelium but loss of VANGL2 is not yet apparent. Both these models suggest

**Fig. 6 ROCK activity enhances apical microtubules but shortens their apicobasal tails. a–d** Wild-type embryos were cultured in vehicle or 10 μm Y27632 (Y27). **a** Representative surface-subtracted images showing β-tubulin staining in vehicle and Y27-treated embryos ($n = 8$ each). **b** Quantification of microtubule apical staining in Y27-treated embryos compared with vehicle controls. **c** Optically-resliced cross-section showing microtubule apicobasal tails (from $n = 6$ embryos each). **d** Quantification of apicobasal tail length in vehicle and Y27-treated embryos. $p$ values from Student's $T$-test. **e** Representative surface-subtracted images showing ROCK1 staining in the apical neuroepithelium of control ($n = 8$) and Cre;Fl/− ($n = 7$) littermate embryos. The Cre;Fl/− embryo image is shown with (right) and without (left) the Vangl2-deleted EGFP cells. Scale bar = 20 μm. **f** Quantification of overall ROCK1 intensity in the apical neuroepithelium of control ($n = 8$) and Cre;Fl/− ($n = 7$) littermate embryos, demonstrating significantly lower signal in the latter. $p$ value by Students $T$-test. **g, h** Quantification of ROCK1 staining intensity selectively, **g** along cell borders or **h** in the apical cap of neuroepithelial cells in control embryos lacking Cre or the indicated cell types in Cre;Fl/− embryos ($n = 7$ per genotype). These graphs are shown on the same scale to emphasise that ROCK1 border staining is much greater than apical cap staining, limiting visualisation of the latter. Points represent average values for cell types from individual embryos, lines link cell types from the same embryo. *$p < 0.05$ by mixed model analysis accounting for repeated measures from the same embryos. **i** High-power image of ROCK1 staining in the apical neuroepithelium shown in Fire LUT. Green border = Vangl2-deleted cell surrounded by Vangl2-replete neighbours. Arrow = ROCK1 localisation at junctions. Scale bar = 10 μm. **j** Schematic illustration of ROCK's action. Refer to Fig. 5 for colour coding. ROCK activity favours apicobasal microtubule tails over their apical networks. ROCK localises more intensely in the apical cap of Vangl2-deleted cells (green), which have diminished apical microtubules but longer tails than their neighbours. All microscopy images were taken using AiryScan.

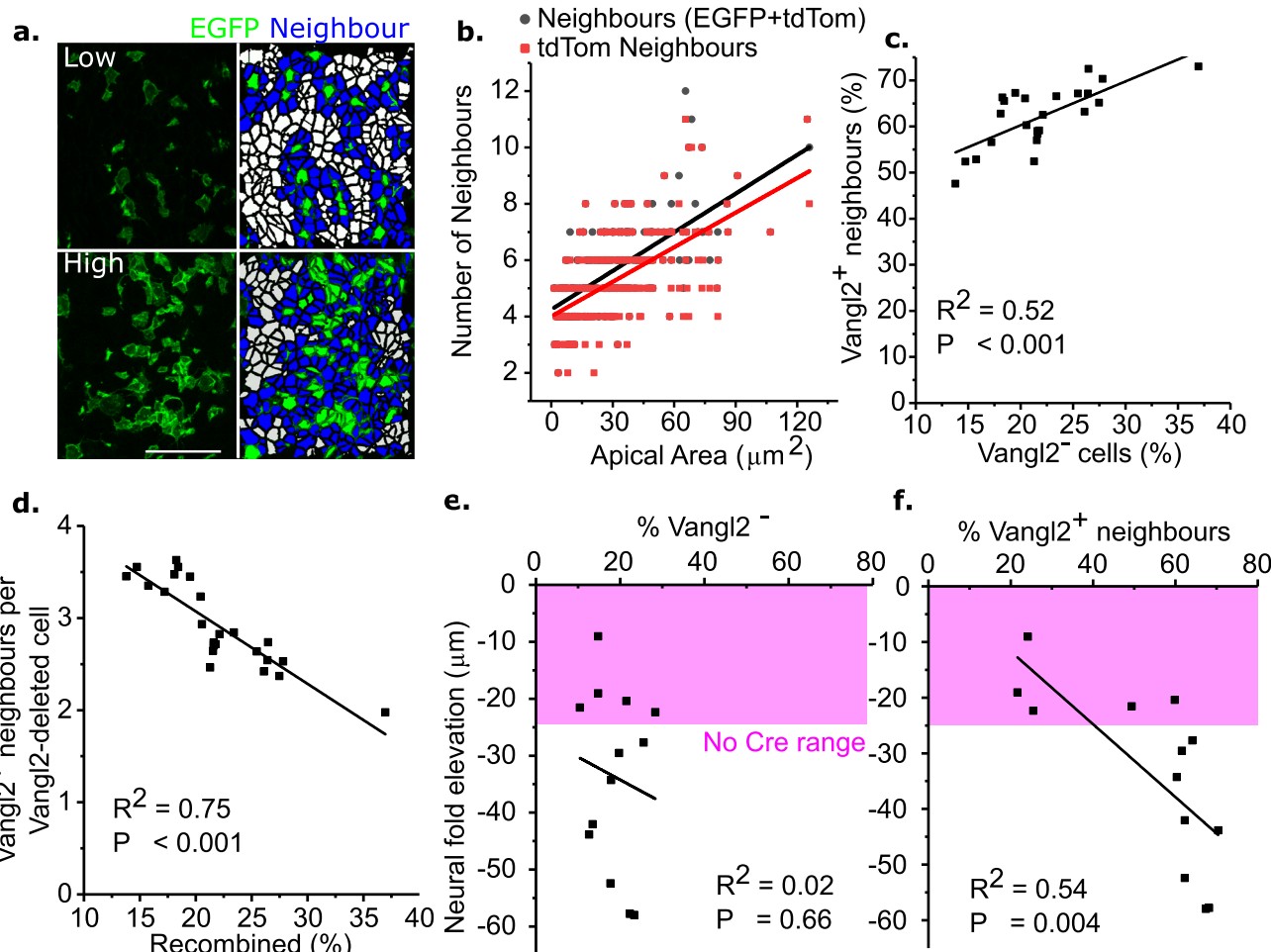

**Fig. 7 The proportion of *Vangl2*-replete cells neighbouring a deleted cell correlates with PNP eversion. a** Illustrative examples of surface-subtracted neuroepithelia with low and high levels of EGFP recombination. Cell borders were determined from ZO1 staining. Green = Vangl2-deleted, blue = neighbour and white = distant cells. Scale bar = 50 μm. **b** Correlation between apical area and the number of neighbours (as predicted by Lewis' law) of Vangl2-deleted cells. EGFP+ cells infrequently neighboured each other so total and Vangl2-replete (tdTom) cells were counted separately. Each dot represents a cell (234 cells from three embryos). **c** Positive correlation between the proportion of neuroepithelial cells which are EGFP+ and the proportion of EGFP− cells which neighbour an EGFP+ cell. Each dot represents an embryo. **d** Inverse correlation between the proportion of unique Vangl2-replete neighbours per Vangl2-deleted cell and overall % Cre-mediated recombination, demonstrating that non-autonomous effect amplification is limited by neighbour-sharing. **e, f** Correlation between neural fold eversion (negative elevation) in Cre;Fl− embryos and the proportion of cells which are Vangl2-deleted (**e**) or neighbours (**f**) in 15–20 somite stage embryos. The magenta box indicates the minimum degree of neural fold elevation observed in No Cre littermates. Each data point represents an individual embryo. $p$ values and linear trend lines from Pearson correlation.

either VANGL2 abundance or activity must fall below a threshold for reduced apical constriction to become detectable. A threshold may be caused by the requirement for an absolute number of VANGL2 membrane complexes to establish PCP signalling, or to aggregate cellular responses dependent on its downstream signalling. This distinction may be important for patients who harbour hypomorphic versus null mutations in this gene[8,17,69], and should be investigated in future work.

The molecular mechanisms downstream of PCP signalling involve dishevelled-dependent recruitment of ROCK and the F-actin polymerising protein dishevelled-associated activator of morphogenesis (DAAM)1[26]. Vangl2 may suppress this contractile machinery because it suppresses dishevelled/DAAM1 interaction[76], potentially explaining why Vangl2-deleted cells apically constrict more than their neighbours. Previous mouse studies suggest aspects of this pathway are conserved in mammals: pharmacological inhibition of ROCK similarly stops apical constriction in mouse embryos, and although Daam1 homozygous deletion does not stop NT closure it interacts genetically with heterozygous Vangl2 mutation to cause spina bifida[77]. However, there are important species differences in this pathway between chick and mouse embryos. Whereas pharmacological ROCK inhibition very dramatically reduces neuroepithelial F-actin intensity in chick embryos[26,78], its inhibition with the same compound expands the region of F-actin apical localisation due to an increase in F-actin versus G-actin proportion in mouse embryos[22,41]. Additionally, whereas the adherens junctions in the anterior chick neuroepithelium are primarily composed of high-affinity E-cadherin[26,79], the mouse PNP neuroepithelium exclusively expresses lower-affinity N-cadherin protein[80,81]. Here, we show that these adherens junction complexes remain apparent in both Vangl2-deleted cells and their neighbours.

Adherens junctions physically couple the contractile actomyosin cytoskeleton between cells, allowing force transmission across epithelia. Two pools of myosin generate apical constriction forces transmitted through these junctions: circumferential contractile networks around the cell cortex and medio-apical networks[29]. Previous studies of myosin activation in the apical neuroepithelium have identified cortical networks in mouse, chick and Xenopus. Phosphorylated, active myosin-II is primarily localised to these cell borders. A large body of work in Xenopus demonstrates that the apical determinant Par3 physically associates with PCP components to localise them to the apical surface[82], where they reside preferentially at contractile cell-cell junctions[83]. Resulting apical constriction of neuroepithelial cells requires the endosomal trafficking protein Rab11[84], whose apical translocation is vangl2-dependant in Xenopus blastopore lip cells[24]. While it has not yet been possible to determine whether these molecular interactions are conserved in mammals, here we demonstrate that the presence of VANGL2 is required for neighbouring cells to recruit myosin-II to their cortex.

In addition to its predominantly cortical localisation, super-resolution imaging also revealed active myosin on the mouse neuroepithelium apical cap. Visualisation of apical cap myosin was substantially improved when total myosin-IIb was visualised. We propose this difference is likely due to simple stoichiometry: the phosphorylated form of the small regulatory light chain presents a far less substantial target for visualisation than the long heavy chain. Alternative explanations may include differential turnover of apical cap myosin, or F-actin cross-linking roles of myosin-II occur independently of its regulatory light chain-dependant motor function[85,86]. Irrespective, the co-occurrence of apical myosin and dense apical microtubules in interphase cells is striking. The actomyosin and microtubule cytoskeletons are physically linked and myosin contractility deforms the microtubule network[87,88]. Here we find that they are also linked in their regulation by ROCK, whose pharmacological inhibition reduces apicobasal tail length but increases the density and homogeneity of the apical network. Equally dramatic increases in microtubule stability following ROCK inhibition have previously been reported in cultured cells[89]. ROCK inhibition may therefore impair apical constriction through both deregulation of actomyosin localisation, as previously documented[22,41], and by expanding the apical microtubule network which must be deformed during constriction.

Apical localisation of PCP components occurs through trafficking in vesicles along apicobasally-oriented microtubules in the zebrafish neuroepithelium[54]. Apicobasally-oriented non-centrosomal microtubule tails have been extensively documented in the Xenopus neuroepithelium[90-92]. The consensus that they biomechanically contribute to NT closure in that model is consistent with our findings. Here we show that Vangl2 deletion non-autonomously shortens microtubule tails in the mouse neuroepithelium, placing VANGL2/ROCK signalling upstream of microtubule organisation. However, whereas the primary function of microtubule tails in the bilayered Xenopus neuroepithelium is believed to be cell elongation[93], shortened tails in the pseudostratified epithelium of Cre;Fl/− embryos were not associated with apical-basal shortening. Our findings are also in contrast to previous studies which implicate microtubule polymerisation in the establishment and maintenance of PCP component polarisation[54,94]. Molecular mechanisms underlying this seemingly bidirectional interaction are a matter for future work, but our findings reveal a dynamic interplay between PCP, microtubules and actomyosin in the mammalian neuroepithelium. It has been suggested that promotion of apical constriction is one of the beneficial consequences of folate, which may contribute to its ability to prevent a proportion of human NT defects[95,96]. These interactions may therefore be specifically disrupted in patients with mosaic PCP mutations and may identify more widely-applicable therapeutic targets.

Patients with non-syndromic NT defects rarely receive a genetic diagnosis relevant to prevention or genetic counselling. Our findings may begin to explain this short-fall. Mutations in neuromesodermal progenitor cells which affect only a small proportion of neuroepithelial cells would not be detectable with diagnostic genotyping methods. These mutations may be predisposed to by agents known to decrease genomic stability and increase NT defect risk, including pollutants[97-99], folate deficiency[100,101], and diabetes[102-105]. Our findings provide a framework to understand the consequences of mosaic mutations on congenital structural malformations.

## Methods

**Animal procedures**. Studies were performed under the UK Animals (Scientific Procedures) Act 1986 and the Medical Research Council's Responsibility in the Use of Animals for Medical Research (1993). Mice were time-mated overnight and the morning a copulation plug was identified was considered E0.5. Five transgenic alleles were used in this manuscript. Rosa26$^{mTmG}$ (mTmG)[106] is a reporter allele which causes all cells to express membrane tdTomato red fluorescent protein, but is recombined by Cre to express membrane EGFP instead. The conditional Vangl2 floxed (Vangl2$^{Fl}$) allele was as previously described[107] and a Vangl2-null allele was generated by crossing to Actin$^{Cre}$, and subsequent breeding out of that ubiquitous Cre more than ten generations before the start of these studies. Three conditional Cre drivers were used in this study. Sox2$^{CreERT2}$ was as previously reported[48]. Nkx1-2$^{CreERT2}$ is a tamoxifen-inducible Cre driver which recombines in axial progenitor cells, persistently lineage-tracing the neuroepithelium[49]. Stud males with Nkx1-2$^{CreERT2/+}$ and Vangl2$^{Fl/−}$ alleles were bred with females harbouring conditional Vangl2 alleles as well as homozygous mTmG reporters. This cross was used to lineage-trace Vangl2-deleted cells at least 24 h after tamoxifen administration. The third conditional Cre driver used was Grhl3$^{Cre}$ [108]. This Cre recombines throughout the surface ectoderm, and in an increasing proportion of neuroepithelial cells as previously observed[14]. Activation of neuroepithelial Grhl3$^{Cre}$ is not temporally restricted, but predictably produces mosaic recombination at later stages of posterior neuropore closure. All mouse colonies were maintained in house on a C57Bl/6 background and bred from 8 weeks of age. Cre-negative littermates generated in the course of colony maintenance were used as WT controls for embryo culture.

Tamoxifen (10 mg/mouse) was administered at 8:00 a.m. on E8 as previously validated[47]. This dose of tamoxifen was halved when collecting foetuses later than E11 as the high dose produced foetotoxic effects in wild-types.

**Embryo collection and culture**. Embryos were harvested around E9-13.5, dissected free of extra-embryonic tissues[109] and rinsed in PBS prior to fixation in 4% paraformaldehyde, pH 7.4, overnight at 4 °C. For tubulin staining, embryos were dissected, washed and fixed in solutions pre-warmed to 37 °C. Yolk sacs were collected from each embryo for genotyping. Mouse whole embryo culture and treatment with the ROCK inhibitor Y27632 (Cambridge Biosciences, SM02-1) was performed using well-established protocols[22,52].

**Whole mount staining**. Primary antibodies were: rabbit anti-VANGL2 (Millipore clone 2G4, as previously validated[74], 1:100 dilution), rabbit anti-MHCIIb (BioLegend PRB-445 and Abcam clone 3H2, 1:200), rabbit anti-Ser19 pMLCII (Cell Signalling Technology #3671, 1:100), rabbit anti-K40 acetylated α-TUBULIN (Abcam EPR16772, 1:200), mouse anti-β TUBULIN (Insight Biotechnology clone AT5B2, 1:200), mouse anti-β-CATENIN (Santa Cruz Biotechnology, clone E-5, 1:100), mouse anti-N-cadherin (Cell Signalling Technology clone 13A9, 1:150), rabbit anti-ZO1 (Thermo Scientific clone 40-2200, 1:100), rabbit anti-ROCK1 (Abcam, ab45171, 1:100) and chicken anti-GFP (Abcam ab13970, 1:300). Alexa Fluor™ 405, 488, 568 and 647-conjugated secondary antibodies and phalloidin were from Thermo Scientific. EGFP was typically detected using endogenous fluorescence, but this was quenched when antigen retrieval was required.

Paraformaldehyde-fixed embryos were permeabilised in PBS with 0.1% Triton X-100 (PBT) for 1 h at room temperature, blocked overnight in a 5% BSA/PBT at 4 °C and incubated overnight in a 1:150 dilution of primary antibody in blocking solution at 4 °C. Embryos were then washed 3 × 1 h at room temperature in blocking solution, incubated for 2 h at room temperature in 1:300 dilution of Alexa Fluor®-conjugated secondary antibodies. Excess secondary antibody was removed by washing for 1 h in blocking solution and a further 2 × 1 h in PBT at room temperature. Visualisation of pMLC-II and N-Cadherin required antigen retrieval. This was performed by pre-incubating embryos in 10 mM sodium citrate acid, 0.05% Tween 20, pH 6.0 for 30 min at room temperature to ensure penetration throughout the embryo, replacing the sodium citrate solution and incubating the embryo at 95 °C for 20 min, then cooling at room temperature for 30 min. In some cases, mild cross-reactivity of the anti-chick secondary was observed, producing faint additional signal in the EGFP track which did not impede identification of EGFP+ cells and was readily corrected by subtracting the original signal.

Images were captured on a Zeiss Examiner LSM880 confocal using a 20×/NA1.0 Plan Apochromat dipping objective. Embryos were typically imaged with X/Y pixel sizes of 0.59 µm and Z-step of 1.0 µm (speed = 8, bidirectional imaging, 1024 × 1024 pixels). Images were processed with Zen2.3 software and visualised as maximum projections (to show intensity) or 3D reconstructions (to show structure) in Fiji.

Myosin and tubulin imaging was performed on a Zeiss Examiner LSM880 confocal using a 20×/NA1.0 Plan Apochromat dipping objective with AiryScan Fast (low zoom) or SR (high zoom) in-built settings.

In order to selectively visualise the apical surface (~3 µm deep into the tissue), our previously-described Surface Subtraction macro was used[14,81]. Images which have undergone this processing are described as being surface-subtracted.

**Morphometric analyses**. PNP length, neural fold elevation and neuroepithelial cross-section analyses were performed in phalloidin wholemount or reflection-imaged PNPs. Reflection mode images were captured using a 633 nm laser and MBS T80/R20 beam filter. To measure neuroepithelial thickness, transverse signal was enhanced post acquisition using sequential local contrast enhancement (CLAHE) in Fiji (block size 50, histogram bins 150, maximum slope 3 then repeated with block size 30, histogram bins 100, maximum slope 3) with 30 pixel rolling ball radius background subtraction before each CLAHE iteration.

Semi-automated cell segmentation was performed with Cellpose[110], which implements a pre-trained U-net neural network to predict intracellular spatial flows (https://github.com/MouseLand/cellpose). One embryo was excluded from further analysis due to poor segmentation. Cells at image edges were excluded by replacing labels within a 30 pixel-wide border of each image with background values. EGFP fluorescence intensity was rescaled to floating point (0–1). A cell was defined as EGFP+ if the intensity of at least 75% of intracellular pixels was greater than 0.08; conversely, a cell was defined as EGFP− if the intensity of at least 75% of intracellular pixels was less than 0.06. Cells were classed as neighbours if the mask of one cell was found to overlap with the dilated mask of another. EGFP negative cells without an EGFP positive neighbour were classed as tdTom distant ($D$), and those with a neighbour were classed as tdTom neighbour ($N$). Cell area ($A$), perimeter ($P$), solidity, and minor and major axis lengths were calculated in scikit-image[111]. All images, outputs, and code used to generate these data is available as a jupyter notebook at https://github.com/timjedwar/Vangl2-cell-morpho.[112]

**Laser ablation and live embryo imaging**. Annular and cable laser ablations were performed using a MaiTai laser (SpectraPhysics Mai Tai eHP DeepSee multiphoton laser). For annular ablations a 30 µm diameter ring was ablated at 710 nm wavelength, 80% laser power, 0.34 µs pixel dwell time, 10 iterations. Cable ablations were

performed using the same laser settings applied to a vertical (y-axis) line ~10 µm long. Immediately prior to ablation, embryos were dissected from the amnion, stained for 5 min at 37 °C in CellMask™ Deep Red (Thermo Fisher, 1:100 dilution in DMEM), positioned in wells cut out of 4% agarose gel in DMEM, submerged in dissection medium and maintained at 37 °C throughout imaging. Microsurgical needles from 11-0 Mersilene (TG140-6, Ethicon) and 10-0 Prolene (BV75-3, Ethicon) were used to hold the PNP pointing upwards. Cluster retraction was analysed by measuring the change in cluster area before and immediately after ablation using peripheral landmarks, such as cell junctions to define cluster boundaries. Cell retraction was analysed by drawing around each cell before and immediately after ablation, then averaging the results for cells of the same type such that the embryo was used as the unit of measure. Illustrative particle image velocimetry (PIV) was performed in Fiji.

Live imaging was carried out using previously-described culture and imaging conditions[13,45]. Endogenous membrane-localised tdTomato or EGFP expressed from the mTmG locus was visualised. Each embryo was positioned by making a small hole through the amnion and yolk sac and exteriorising the PNP to allow visualisation. A 10-0 Prolene microsurgical needle was curved through the allantois into agarose to keep the PNP pointing upwards and other needles were used to stop the yolk sac drifting onto it. The heart beat continued throughout imaging and the PNP commonly drifted as the embryo grew/deformed. Z-stacks of the neuroepithelium were captured once per minute and the field of view was recentred after every two or three stacks. The resulting stacks were concatenated, processed with five iterations of Richardson-Lucy deconvolution using DeconvolutionLab, and 3D-registered in Fiji. Only cells which were present in all imaged time points were analysed, in each case by manually identifying the last Z-slice in which the cell's apical surface was visible and drawing around it sequentially at every time point. While this is far more laborious than the commonly-used approach of selecting an optical cross-section for analysis, it is essential because the tissue curves during progression of closure. When analysing PNPs with recombined cells, for each EGFP+ cell a neighbour and a distant cell with similar initial morphology were selected for annotation. Cells which divided during imaging were not analysed.

**Statistical analysis and reproducibility**. Comparisons between two groups were by Student's *t*-test accounting for homogeneity of variance in Excel or in SPSS (IBM Statistics 22). Comparison of multiple groups was by one-way ANOVA with post-hoc Bonferroni in OriginPro 2017 (Origin Labs), which also tests normality and homogeneity of variance. Multivariate analysis was by linear mixed models in SPSS accounting for repeated measured from individual cells and embryos, and for multiple testing with a posthoc Bonferroni. Graphs were made in OriginPro 2016 (Origin Labs) and are represented as box plots or as the mean ± SEM when several groups are shown per measurement level. Box plots show the median, interquartile range and 95% CI.

Sample sizes for morphometric and laser ablation experiments were based on previous studies. A pilot study of quantifying apical constriction is described in the results. Blinding to CreERT2 positivity was generally not possible, but analyses were carried out without knowing whether embryos were Cre;Fl/Fl (no spina bifida) or Cre; Fl/-. Thus, analyses were generally blinded to *Vangl2* deletion status and no embryos were excluded after analysis. There were three exceptions to this blinding. The first exception is when VANGL2 itself was visualised given loss of VANGL2 signal was obvious in Cre;Fl/− embryos only; these data are analysed quantitatively. The second exception was when selecting embryos for live imaging. In order to ensure that control and experimental embryos were imaged, GLG inspected neural fold eversion, which by then was a recognised feature of Cre;Fl/− embryos (note that this is only possible when the whole PNP can be seen, not when processing zoomed images of the apical neuroepithelium). This was only mis-judged in one embryo. The third exception was AiryScan imaging of myosin and tubulin, for which only experimental embryos could be imaged due to processing limitations. To circumvent this, when quantifying tubulin tail length each tail was saved as a separate image with a blinded key indicating whether it was EGFP+ or not.

$p < 0.05$ was considered statistically significant and all tests were two-tailed. All images are representative of observations in at least three embryos from independent litters.

**Reporting summary**. Further information on research design is available in the Nature Research Reporting Summary linked to this article.

## Data availability

The authors declare that all data supporting the findings of this study are available within the article and its supplementary information files or from the corresponding author upon reasonable request. Source data are provided with this paper as both the values used to generate each graph and SVG images at original resolution available through the Mendeley data repository[113]. All reagents are commercially available. Source data are provided with this paper.

## Code availability

Cellpose processing is available on GitHub https://github.com/timjedwar/Vangl2-cell-morpho.[112] The in-house Fiji macro developed to identify and extract the surface of 3D structures is available from https://github.com/DaleMoulding/Fiji-Macros and related resources are available from the corresponding author on reasonable request or previously published[14].

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

## Acknowledgements

This study was funded partly by a Wellcome Trust Postdoctoral Clinical Research Training Fellowship (107474/Z/15/Z) and partly by a Wellcome Clinical Research Career Development Fellowship (211112/Z/18/Z), both to G.L.G. A.R.M. was supported by a Child Health Research CIO award (to N.D.E.G.). A.J.C. and N.D.E.G. are supported by Great Ormond Street Hospital Children's Charity. Research infrastructure within the

institute is supported by the NIHR Great Ormond Street Hospital Biomedical Research Centre. The views expressed are those of the authors and not necessarily those of the NHS, the NIHR or the Department of Health. We thank Dr Dale Moulding for his help with image processing, as well as Dawn Savery and the biological services staff for help with transgenic colonies.

## Author contributions

Conceptualization: G.L.G.; Methodology: G.L.G., E.M., A.R.M., T.J.E.; Formal analysis: G.L.G., E.M., A.R.M., T.J.E. and I.A.; Investigation: G.L.G., E.M., A.R.M.; Resources: G.L.G., N.D.E.G., A.J.C.; Data curation: G.L.G., T.J.E.; Writing—original draft: G.L.G.; Writing—review & editing: all authors; Figure creation: G.L.G., T.J.E.; Supervision: G.L.G., N.D.E.G., A.J.C.; Project administration: G.L.G.; Funding acquisition: G.L.G., N.D.E.G., A.J.C.

## Competing interests

A.J.C. acts as paid consultant for ViiV Healthcare Limited, with fees going to support his research programme. G.L.G, E.M., T.J.E, A.R.M, I.M. and N.D.E.G declare no competing interests.
