## [Peer Review File · Nature Communications]

Reviewers' Comments:

Reviewer #1:

Remarks to the Author:

Manuscript ID: NCOMMS-20-25036

Title: Cell non-autonomy amplifies disruption of neurulation by mosaic Vangl2 deletion.

Authors: Gabriel L. Galea, Eirini Maniou, Abigail R. Marsall, Nick DE Greene, and Andrew J. Copp

Overview:

In this manuscript, the authors were trying to answer one of the more interesting questions in the neural tube defect field, by studying the effects of a mosaic neuroepithelial Vangl2 deletion on posterior neuropore closure and probing the underlying mechanisms. As Vangl2 is associated with the pathogenicity of spinal NTDs in both human and mice, identifying autonomous and non-autonomous role of Vangl2 gene during neural tube closure are an interesting study to better understand the etiology of NTDs. More importantly, there are the increasing needs to know the somatic mutation of causative genes in NTDs due to the nature of its sporadic and genetic mosaicism. The authors are trying to answer how Vangl2 mosaicism in neuroepithelium could cause morphological disruption (apical constriction disruption) during spinal neurulation utilizing Vangl2 mouse models along with lineage tracing and live embryo imaging technologies.

Using an oral tamoxifen inducible neuromesodermal progenitor marker *Nkx1.2CreERT2* and *Rosa26mTmG* reporter strain, the investigators generated *Nkx1.2CreERT2; Vangl2^{Fl}/-*; *Rosa26mTmG/+* embryos to trace the Vangl2 expression (GFP-) or mosaic Vangl2-deletion neuroepithelial cells (GFP+) with GFP signaling. The investigators found that mosaic neuroepithelial Vangl2 deletion could prevent neural fold elevation which leads to the failure of posterior neuropore (PNP) closure, producing pre-spina bifida lesions. The underlying mechanism is that Vangl2-deleted cells reduced myosin-II recruitment to their neighboring cell borders, as well as diminishing tubulin tail length, which compromised neuroepithelial apical constriction, an important event during PNP. The authors observed that only a 16% difference in Vangl2 deletion in the neuroepithelium was sufficient to compromise neural fold elevation during spinal neurulation. This is possibly the result of behavioral changes in the apical constriction of neighboring cells that remain Vangl2 replete. The microtubules and actomyosin cytoskeleton are involved in the non-autonomous Vangl2 mediated regulation of apical constriction in neuroepithelial cells during spinal neurulation. This study provides a model to explain the contribution of post-zygotic mutations to the etiology of human neural tube defects, which was recently reported by a group of US and Chinese scientists.

Strengths of the manuscript:

1. The manuscript provides a possible answer for a very interesting question that can direct conceptual improvements in our thinking about the events in neural tube closure.
2. The methodologies (such as lineage tracing and live embryo imaging) that the investigators used to prove their hypothesis are robust and relevant to testing their hypothesis.
3. The authors used appropriate the statistical analyses.

Major Concerns:

1. The authors used a Vangl2 conditional mouse along with a *Nkx1.2CreERT2/+* line. The phenotypes between *Cre;F/F* and *Cre;F/-* are quite different as one line had 54% pre-spina bifida lesions while the other did not. There is still some portion of Vangl2 reduction in *Cre;F/F* embryos (Fig 1D) although the reduction seems much milder than *Cre;F/-* embryos, as the authors clearly articulated. This raised the question of whether there is any Vangl2 reduction threshold to induce cellular mosaicism that results in the change of apical constriction in neighboring cells. To clarify this, the comparison of Vangl2 expression between *Cre;F/F* and *Cre;F/-* should be added in Fig. 1F. Or the Cre recombines and the cells are scattered among the neuroepithelium in the *Grl3-Cre* mouse line, although it also recombines in the surface ectoderm. I am curious whether or not the

authors observed a similar apical constriction change in neighboring cells of the Vangl2 depleted cells in neuroepithelium with the Grhl3-Cre lines?

2. In Fig 3 of the manuscript, the investigators delineate the cell autonomous effect vs cell non-autonomous effect of apical construction changes of neighboring cells of Vangl2 depleted cells in by measuring retraction rate of tdTom+ vs EGFP+ cells. If there are any neighboring cells (Vangl2 replete cells) near two or more successive Vangl2 depleted cells in a row, the retraction (Fig. 3) or constriction (Fig. 4) rate of the Vangl2 replete cells is different from a cell that is close to only one Vangl2 depleted cells? Do the investigators know anything about retraction or constriction of Vangl2 depleted cell immediately next to a Vangl2 depleted cell?

3. Cell adhesion changes could be one of major reasons underlying the differential retraction/constriction caused by Vangl2 depletion as the authors described. Although it turned out that this is not the case with the Vangl2 mosaic effect, the addition of the statistical data in adherens and tight junction staining in Sup Fig 4 is important to be highlighted so that it strengthens the results of the author's conclusions.

4. This reviewer is wondering if this cell non-autonomous effect of Vangl2 mosaic depletion in the neuroepithelium is relevant to all other PCP genes that regulate apical constriction, or if this effect is only specific to Vangl2. It will be helpful to add any comments on this issue in the Discussion.

Additional Concerns:

1. The main idea driving this study, that somatic Vangl2 deletion causes a failure of PNP closure and results in (pre-)spina bifida, was previously reported by the senior authors in 2018 in the journal *Disease Models and Mechanisms* (PMID: 29590636) journal. In their previous study, they used Grhl3cre to delete Vangl2 in the surface ectoderm (SE), whereas in the present study the investigators now use Nkx1.2CreERT2; Vangl2^{Fl/-}; Rosa26mTmG/+ embryos which had a more restricted portion of Vangl2-deleted in the neuroepithelial cells. Most of the assays, including apical constriction, cell polarity and cytoskeleton changes are similar between the two studies. Although, both studies were performed very well and provided solid data to support their conclusions, the novelty for this study is limited, especially in light of the impact factor of this journal.

2. On Page 11, lines 9-10, the authors claimed that Vangl2-deletion in 16% percent of the neuroepithelial cells is sufficient to cause spina bifida. There is no data showing a spina bifida phenotype, as the results presented were pre-spina bifida. Readers may wonder whether any of the mice with somatic Vangl2 deletion (Nkx1.2CreERT2; Vangl2^{Fl/-}; Rosa26mTmG/+) were allowed to be liveborn and presented with a spina bifida phenotype, and what was the prevalence of any spina bifida affected newborns. The lack of this information is a shortcoming of the manuscript.

3. On Page 14, lines 8-9, the authors indicated that "Vangl2 deletion non-autonomously shortens microtubules tails in the mouse neuroepithelium placing Vangl2 upstream of microtubule organization." It would be helpful if the authors provided additional details explaining just how the Vangl2 deletion non-autonomously regulated microtubule tails length puts Vangl2 upstream of microtubule organization.

4. The underlying mechanism by which somatic Vangl2 deletion regulates microtubule and cytoskeleton is not well studied. Single cell RNAseq analysis which can trace cell lineage and cluster Vangl2-replete and Vangl2-delete cells should be able to provide more information on the Vangl2 somatic deletion effects on gene expression. While not integral to the manuscript, the authors might consider including scRNA seq in any revision.

5. There appears to be an important omission in the author's introductory literature review. The first paper establishing the association between VANGL2 and human neural tube defects (*N Engl J Med.* 2010. PMID: 20558380) should be included in the references cited.

Reviewer #2:

Remarks to the Author:

The manuscript by Galea et al. investigates the hypothesis that genetic mosaicism of PCP signalling during post-zygotic embryonic development is sufficient to cause NT closure defects. The authors model mosaic loss using Tamoxifen induced Nkx1.2 driven deletion of Van Gogh-like (Vangl)2 and examine the effect on spinal neurulation. The manuscript shows that Vangl2 deficient cells induce a cell non-autonomous failure of neighbouring neuroepithelial cells to undergo apical constriction, using elegant live-imaging and ablation approaches. This explains why loss of Vangl2 in only a small percentage of neuroepithelial cells can have deleterious effects on the process of neural tube closure.

I suggest that the manuscript is acceptable for publication if minor comments below are addressed.

1. Figure 1 establishes the model that the authors use for mosaic Vangl2 deletion. It is unclear to me what is shown in panel C. I suggest that the authors add a schematic of the embryo and the structures that are referred to in panel C. Also a schematic showing the steps of NT closure across the developmental timepoints described in the manuscript would be useful.

2. Cre;Fl/- cellular orientation in Figure 2B is not randomised, instead there appears to be a preference for a RC direction in the Vangl2 Fl/-. Can the authors explain this effect?

3. In Figure 3, the authors use laser ablation to show that clusters of cells that consist of Vangl2 deficient cells and their GFP negative neighbours are under reduced apical tension and that this effect is due to non-cell autonomous changes in tension in the neighbours. It would be interesting if the authors could re-analyse the mixed clusters to examine if there is a direct correlation between the GFP+/GFP- ratio and retraction range upon ablation? If a cluster consists of only one GFP+ cell you would expect a greater loss in retraction compared to a cluster that is mostly GFP+?

4. Is it also possible that neighbouring GFP negative cells are actively extruding the Vangl2 deficient cell which results in enhanced cap MHC-II and longer microtubule tails? This would likely also result in overall smaller Vangl2 deficient cells and larger neighbours, as shown in Figure 2?

Reviewer #3:

Remarks to the Author:

In this study, Galea et al. show that mosaic deletion of the PCP pathway member Vangl2 within the neuroepithelium is sufficient to impair posterior neuropore closure, resulting in neural tube defects. Vangl2-deleted cells are able to undergo apical constriction, but their neighbours fail to constrict as a result of reduced myosin II accumulation and shortening of microtubule tails. They propose that this cell non-autonomous mechanism between Vangl2-deleted cells and their multiple cell neighbours amplifies the neurulation defects, explaining why as little as 16% deletion of Vangl2 within cells of the neuroepithelium is sufficient to block neural tube closure. These findings are important in understanding how mosaic patterns of post-zygotic mutations within developing tissues can produce disease phenotypes.

However, the study does not identify a clear mechanistic link between Vangl2 and its downstream effects on the cytoskeleton and apical constriction. For example, how does Vangl2 affect myosin localization and microtubule organization? Does Vangl2 regulate myosin and microtubules via independent or common pathways?

Secondly, it remains unclear how Vangl2-deleted cells can affect apical constriction of neighboring

cells, while retaining the ability to constrict themselves. How does PCP activity change in the cells adjacent to Vangl2-deleted cells? If PCP signaling is altered in both types of cells, what regulates the differing apical constriction outcomes in these cells? Perhaps the authors could check previous work in other *in vivo* systems showing that disruption of apical constriction in one cell affects the ability of neighbouring cells to constrict (see for instance Samarage et al Dev Cell).

Other comments:

- Although Vangl2 is indeed a core component of the PCP signaling pathway, can the authors better demonstrate whether the aberrant phenotypes observed in the mosaic Vangl2-deleted neuroepithelium directly result from impaired PCP signaling, or if they stem from other PCP-independent functions of Vangl2?

- Do the Vangl2-deleted cell neighbours undergo a corresponding expansion of their apical surfaces when the Vangl2-deleted cells constrict?

- The analysis of the three types of cell junctions (EGFP/neighbour, neighbour/neighbour, and distant/distant) for MHC-IIb levels could be supplemented with laser ablation experiments to confirm the differences in junctional tension.

- In Vangl2-deleted cells, MHC-IIb is primarily cortical (instead of being enriched at the apical cap). Do the total MHC-IIb levels differ between Vangl2-deleted and control cells, or are the levels unchanged and only the subcellular distribution of MHC-IIb is altered?

Reviewer #4:

Remarks to the Author:

In this very interesting paper, the authors use a clever new approach to mosaic conditional knock-out in the neural plate. They disrupt Vangl2 specifically in a subset of cells, and observe not only cell-autonomous defects, but also non-autonomous non-autonomous defects in the neural epithelium as well as tissue-level neural tube defects. These defects are carefully quantified and related to changes in actomyosin-based cell mechanics and the microtubule cytoskeleton.

In addition to providing new insights into the function of Vangl2, a crucial paradigm for understanding the genetics of neural tube closure, the authors argue that this system may model sporadic somatic mutations in humans and how those could in turn contribute to the sporadic nature of neural tube defects. While it is difficult to bridge this gap in knowledge (as they readily admit), I found this to be a thought-provoking piece of science that I recommend for publication. The main shortcomings are slightly inadequate description and discussion of their genetic system, which is very complex. The description of "sarcomere-like" assembly of myosin at the apical surface of cells has little merit here and should be omitted, opting instead for language that implies less about the very-unclear structure of actomyosin at that domain.

Specific comments:

1. Page 3, lines 15-20: The citations here are useful and interesting but come from a variety of animals and tissues. I'd appreciate clarity on the systems these phenomena occur in as none of them are neural tube closure and that isn't made clear.

2. Page 4, lines 26-27: Is it known or is there a hypothesis why Grhl3-Cre drives patchy recombination in the neural epithelium? Is Cre expressed at low levels across the neural epithelium or is the expression of Cre itself patchy?

3. Figure 1A/B: would appreciate if the anteroposterior axis was labeled.

Figure 1G: It is unclear how EGFP expression was compared between the two embryos. Given the one on the left closed, was EGFP expression in the neural epithelium inferred from the surface epithelium? Do you have evidence that surface and neural recombination levels correlate?

4. Figure 2: I would like to see a comparison of the shape of the Cre;Fl^{-/-}, EGFP⁺ to the other conditions, perhaps via aspect ratio. Also, "EGF" should be replaced by "EGFP"
5. Figure 3: Would like to see EGFP^{+/-} arranged in the same order as previous figures
6. Figure 4: 4B is confusing but I think I get it...
7. Page 7/Methods/Supplementary Figure 5: Would like to see mention somewhere of what imaging marker was used in the live-imaging experiments.
8. Page 9, lines 6-19: the myosin arrangement could possibly be "sarcomere-like" but I think it is premature to describe it as such based on the presented data. The cited results from *Drosophila* showed a) a fairly different localization pattern with radialized cell polarity at the apical surface and b) more data to demonstrate a "sarcomere-like" actomyosin structure at the apical surface.
9. Page 11, lines 11-25: This is useful information but I wonder if it wouldn't belong better at the beginning of the results section, as it addresses technical information that affects the results and generated the concept. In particular, I think the point that Grhl3-Cre induces Vangl2 recombination relatively late in neural tube closure is a point that should be made clearer early on.

We thank the reviewers for their constructive comments, which have helped us improve our manuscript. We have undertaken additional experiments and provide four new figures showing their results, in addition to changes to the manuscript text, to fully address all comments.

REVIEWER COMMENTS

Reviewer #1 (Remarks to the Author):

Overview:

Strengths of the manuscript:

1. The manuscript provides a possible answer for a very interesting question that can direct conceptual improvements in our thinking about the events in neural tube closure.
2. The methodologies (such as lineage tracing and live embryo imaging) that the investigators used to prove their hypothesis are robust and relevant to testing their hypothesis.
3. The authors used appropriate the statistical analyses.

Major Concerns:

1. The authors used a *Vangl2* conditional mouse along with a *Nkx1.2CreERT2/+* line. The phenotypes between *Cre;F/F* and *Cre;F/-* are quite different as one line had 54% pre-spina bifida lesions while the other did not. There is still some portion of *Vangl2* reduction in *Cre;F/F* embryos (Fig 1D) although the reduction seems much milder than *Cre;F/-* embryos, as the authors clearly articulated. This raised the question of whether there is any *Vangl2* reduction threshold to induce cellular mosaicism that results in the change of apical constriction in neighboring cells. To clarify this, the comparison of *Vangl2* expression between *Cre;F/F* and *Cre;F/-* should be added in Fig. 1F. Or the *Cre* recombines and the cells are scattered among the neuroepithelium in the *Grhl3-Cre* mouse line, although it also recombines in the surface ectoderm. I am curious whether or not the authors observed a similar apical constriction change in neighboring cells of the *Vangl2* depleted cells in neuroepithelium with the *Grhl3-Cre* lines?

We have now undertaken new experiments to confirm that neuroepithelial apical constriction also decreases when *Vangl2* is deleted using *Grhl3^{Cre}*. This *Cre* also allowed us to confirm the reviewer's proposal that a threshold of *Vangl2* deletion is required to diminish constriction. *Grhl3^{Cre}* is a non-inducible *Cre* which we previously reported begins to lineage-trace neuroepithelial cells at a low level as early as the 8-somite stage then increases during development (Galea et al, Disease Mod Mech, 2019). However, these EGFP lineage-traced cells will still have VANGL2 protein and loss of VANGL2 is not visible until the ~19-somite stage (Supplementary Figure 4A-C, shown below). This non-inducible *Cre* therefore cannot be used to identify cells which have lost VANGL2 in

the same way as temporally-controlled activation of *Nkx1.2^{CreERT2}* does. Instead, it provides a system in which VANGL2 is selectively lost at later stages of development. This is now explained in the results:

“To corroborate this finding in an independent model, we used *Ghr13^{Cre}* to mosaically delete neuroepithelial *Vangl2*. *Ghr13^{Cre}* deletion of *Vangl2^{Fl/Fl}* produces late deletion of neuroepithelial VANGL2 and distal spina bifida, without requiring pre-deletion of a *Vangl2* allele¹. Neuroepithelial expression of the *Grhl3* gene is very low at E8.5, when it is largely restricted to the surface ectoderm overlying the neural folds, but increases at later stages². Consistent with this, the knock-in *Grhl3^{Cre}* lineage-traces a small proportion of neuroepithelial cells as early as the 8-somite stage and this proportion increases during development^{1,3,4}. In agreement with this, *Grhl3^{Cre/+}Vangl2^{Fl/Fl}* show patchy loss of neuroepithelial VANGL2 protein at later stages of development, beyond the ~19-somite stage, but not earlier (Supplementary Figure 4A-C).”

We performed annular laser ablations in *Grhl3^{Cre/+}Vangl2^{Fl/Fl}* embryos at the 13-17 somite stages, before VANGL2 protein is visibly diminished, and at the 19-23 somite stages when VANGL2 is visibly lost in patches across the epithelium. Cluster retraction following ablation was not significantly different between *Grhl3^{Cre/+}Vangl2^{Fl/Fl}* embryos and littermate controls at the earlier stages, but was significantly impaired at later stages beyond the VANGL2 deletion threshold. This is now included in the results:

“Neuroepithelial annular ablations performed at earlier developmental stages, before VANGL2 protein is diminished, showed no difference in apical tension between *Grhl3^{Cre/+}Vangl2^{Fl/Fl}* embryos and their *Vangl2*-replete littermates (Supplementary Figure 4D). In contrast, at later developmental stages when VANGL2 is predictably diminished, *Grhl3^{Cre/+}Vangl2^{Fl/Fl}* embryos underwent less retraction following neuroepithelial annular ablation than their *Vangl2*-replete littermates (Supplementary Figure 4D).”

In addition, the *Grhl3^{Cre}* model gave us the opportunity to test whether *Vangl2* deletion diminishes actomyosin contractility in other cell types. We previously identified long actomyosin cables in surface ectoderm cells which biomechanically couple the most recently-fused zippering point caudally to the open neuropore (Galea et al, PNAS 2017). These cables are still present in *Grhl3^{Cre/+}Vangl2^{Fl/Fl}* embryos (Galea et al, Disease Mod Mech, 2019). We therefore ablated them in independent embryos, finding that loss of VANGL2 does not reduce cable tension (Supp Fig.XX E-G). This is now described in the results:

“Additionally, *Grhl3^{Cre}* also deletes *Vangl2* in the surface ectoderm cells surrounding the open PNP¹. These cells assemble long actomyosin cables which biomechanically couple the recently-fused zippering point to the caudal, open PNP^{1,5} (Supplementary Figure 4E). We previously confirmed that these cables remain present in *Grhl3^{Cre/+}Vangl2^{Fl/Fl}* embryos¹. In the current study, we performed cable laser ablations to test whether loss of VANGL2 diminishes contractility of these structures as well (Supplementary Figure 4F). Cable recoil following ablation was equivalent between *Grhl3^{Cre/+}Vangl2^{Fl/Fl}* embryos

and their *Vangl2*-replete littermates (Supplementary Figure 4G), suggesting that loss of VANGL2 reduces contractility selectively in the apical neuroepithelium.”

Supplementary Figure 4: Apical neuroepithelial tension is specifically diminished by mosaic *Vangl2* deletion using *Grhl3^{Cre}*. All panels in this figure show data using the non-inducible *Grhl3^{Cre}*, not *Nkx1.2^{CreERT2}* used throughout the remainder of the manuscript.

- A) As previously reported¹, *Grhl3*^{Cre} produces loss of VANGL2 protein in surface ectoderm (arrows) early in development, but neuroepithelial VANGL2 is only diminished at later stages. Scale bar = 100 μ m.
- B) Surface-subtracted images of somite stage-matched control and *Grhl3*^{Cre/+} *Vangl2*^{Fl/Fl} embryos showing a mosaic pattern of VANGL2 loss in the apical neuroepithelium of the latter. Scale bar = 50 μ m.
- C) Low magnification image to show the location of the neuroepithelial cells in B. Scale bar = 50 μ m.
- D) Quantification of the change in cluster area following annular laser ablation of the apical neuroepithelium. Ablation experiments were carried out at two developmental stages: 13-17 somite stages before VANGL2 protein depletion is apparent and 19-23 somite stage after loss of VANGL2 becomes visible. Cluster retraction is equivalent between controls at both stages of development and is selectively diminished in embryos which have lost VANGL2. Ablations are assumed to include *Vangl2*-deleted cells (the vast majority of ablation annuli would include at least one lineage-traced cell). Points represent individual embryos.
- E) Representative *Grhl3*^{Cre} lineage tracing using the *mTmG* reporter showing near-ubiquitous recombination in the surface ectoderm which assembles long F-actin cables (arrows in inset). Also note the mosaic recombination scattered throughout the neuroepithelium at the 15 somite stage. Scale bar = 100 μ m.
- F) Representative ablation of the cell borders which assemble the F-actin cables showing recoil perpendicular to the ablation (red line). Scale bar = 20 μ m.
- G) Quantification of the change in cell border length immediately following cable ablation of 20-26 somite stage control and littermate *Grhl3*^{Cre/+} *Vangl2*^{Fl/Fl} embryos. Points represent individual embryos. There is no significant difference in length change between genotypes.

As the reviewer states, the *Nkx1.2*^{CreERT2} model produces significant VANGL2 deletion in lineage-traced cells of both Cre;Fl/- and Cre;Fl/Fl embryos, but the level of *Vangl2* expression is reduced significantly more in the former. This data is shown in Figure 1D-F. Lineage-traced cells are scattered throughout the neuroepithelium and are present at variable, low frequencies, such that quantification of absolute protein levels is not currently possible. In conjunction with the *Grhl3*^{Cre} model described above they nonetheless corroborate the reviewers' suggestion that a VANGL2-deletion threshold must be exceeded before neuroepithelial apical constriction is impaired. This is now discussed:

“Neuroepithelial apical tension is largely unchanged in *Nkx1.2*^{CreERT2/+} *Vangl2*^{Fl/Fl} embryos despite mild yet significant reduction in VANGL2 immunolocalisation. Similarly, *Grhl3*^{Cre/+} *Vangl2*^{Fl/Fl} embryos retain neuroepithelial apical tension at earlier developmental stages when this non-inducible Cre abundantly lineage-traces the neuroepithelium but loss of VANGL2 is not yet apparent. Both these models suggest either VANGL2 abundance or activity must fall below a threshold for reduced apical constriction to become detectable. A threshold may be caused by the requirement for an absolute number of VANGL2 membrane complexes to establish PCP signalling, or to aggregate cellular responses dependent on its downstream signalling. This distinction

may be important for patients who harbour hypomorphic versus null mutations in this gene⁶⁻⁸, and should be investigated in future work.”

2. In Fig 3 of the manuscript, the investigators delineate the cell autonomous effect vs cell non-autonomous effect of apical constriction changes of neighboring cells of Vangl2 depleted cells in by measuring retraction rate of tdTom+ vs EGFP+ cells. If there are any neighboring cells (Vangl2 replete cells) near two or more successive Vangl2 depleted cells in a row, the retraction (Fig. 3) or constriction (Fig. 4) rate of the Vangl2 replete cells is different from a cell that is close to only one Vangl2 depleted cells? Do the investigators know anything about retraction or constriction of Vangl2 depleted cell immediately next to a Vangl2 depleted cell?

We have now extended our annular laser ablation analysis and find there is no correlation between the number of *Vangl2*-deleted EGFP+ cells contacting each cluster and the overall retraction (Supp Fig 3B-C). This analysis also confirms that nearly all clusters involved multiple *Vangl2*-deleted cells, demonstrating that these cells are in close proximity to other deleted cells, and yet still retract.

Supplementary Figure 3: Neuroepithelial apical retraction after annular ablation is unaffected in Cre;Fl/Fl embryos.

A. Quantification of the retraction (% change in area) of cell clusters within the ablated annulus. Dots represent a cluster in an individual embryo; each embryo was ablated once only. Control embryos are compared to $Nkx1.2^{CreERT/+} Vangl2^{Fl/Fl}$ (Cre;Fl/Fl) embryos.

B. Quantification of the retraction (% change in area) of EGFP+ and tdTom+ cells within the ablated annulus of Cre;Fl/- embryos graphed against the number of EGFP+ cells contacting the annulus. EGFP+ cells were counted even if they were themselves cut during ablation and not part of the analysed cluster. Each point represents EGFP+ or tdTom+ cells from an individual embryos.

C. Representative ablations showing clusters with two and five EGFP cells.

D. Quantification of the retraction (% change in area) of individual cells within the ablated annulus.

Control embryos are the same as in Figure 3. Points represent individual embryos.

In order to further analyse the potential for a “dose-response” with the number of *Vangl2*-deleted cells each *Vangl2*-replete cell contacts, we analysed apical area of a large number of cells. We confirmed that *Vangl2*-deleted cells have smaller (more constricted) apical areas than their *Vangl2*-replete neighbours, but found no correlation between the apical area of *Vangl2*-replete neighbours and the number of *Vangl2*-deleted cells they contact (Supplementary Figure 5C). This suggests an “all-or-none” non-autonomous mechanism.

Supplementary Figure 5: *Vangl2*-deleted cells have more constricted apical areas than their *Vangl2*-replete neighbouring cells.

A. Schematic representation of the automated image segmentation process using Cellpose to identify cell borders, attribute EGFP status based on intensity cut-offs (see Methods) and identify the number of EGFP+ cells each cell contacts.

B. Quantification of apical areas between the cell types indicated, showing EGFP+ cells (*Vangl2* deleted) are significantly smaller than their fully *Vangl2*-replete neighbours in both the *Cre;Fl/Fl* and *Cre;Fl/-* embryos analysed. Points represent average values for cell types from individual embryos, lines link cell types from the same embryo. * $p < 0.05$ by mixed model analysis accounting for repeated measures from the same embryos.

C. Apical area of individual *Vangl2*-replete cells contacting single or multiple *Vangl2*-deleted (EGFP) neighbours. Points represent individual cells from five embryos.

D-F. Comparison of the standard morphometric parameters: D) circularity, E) shape index and F) aspect ratio, showing no significant differences in these parameters between apical surfaces of the cell types analysed.

3. Cell adhesion changes could be one of major reasons underlying the differential retraction/constriction caused by Vangl2 depletion as the authors described. Although it turned out that this is not the case with the Vangl2 mosaic effect, the addition of the statistical data in adherens and tight junction staining in Sup Fig 4 is important to be highlighted so that it strengthens the results of the author's conclusions.

These data have now been added to the manuscript, confirming no differences between adherens and tight junction localisation between *Vangl2*-deleted and replete cells (Supplementary Figure 6).

4. This reviewer is wondering if this cell non-autonomous effect of Vangl2 mosaic depletion in the neuroepithelium is relevant to all other PCP genes that regulate apical constriction, or if this effect is only specific to Vangl2. It will be helpful to add any comments on this issue in the Discussion.

We thank the reviewer for raising this question, which we intend to pursue in future work. Human cases suggest mutation of other membrane co-receptors may produce similar effects, as we now discuss (p 13):

“Somatic mutations in other PCP membrane complex components, including *FZD6* and *CELSRI*, have also been identified in individuals who have spina bifida⁹, suggesting that the effects documented here may extend beyond *VANGL2*.”

Additional Concerns:

1. The main idea driving this study, that somatic Vangl2 deletion causes a failure of PNP closure and results in (pre-)spina bifida, was previously reported by the senior authors in 2018 in the journal Disease Models and Mechanisms (PMID: 29590636) journal. In their previous study, they used Grhl3cre to delete Vangl2 in the surface ectoderm (SE), whereas in the present study the investigators now use Nkx1.2CreERT2; Vangl2^{Fl/-}; Rosa26mTmG/+ embryos which had a more restricted portion of Vangl2-deleted in the neuroepithelial cells. Most of the assays, including apical constriction, cell polarity and cytoskeleton changes are similar between the two studies. Although, both studies were performed very well and provided solid data to support their conclusions, the novelty for this study is limited, especially in light of the impact factor of this journal.

We are pleased that the reviewer agrees both these studies were performed very well and provided solid data. The non-autonomous effect identified in this study is entirely new and we had no inkling of it when we published the Disease Models and Mechanisms

paper. This is because *Grhl3^{Cre}* cannot be used to lineage-trace *Vangl2*-deleted cells in the same temporally-controlled way as *Nkx1.2^{CreERT2}*.

2. On Page 11, lines 9-10, the authors claimed that *Vangl2*-deletion in 16% percent of the neuroepithelial cells is sufficient to cause spina bifida. There is no data showing a spina bifida phenotype, as the results presented were pre-spina bifida. Readers may wonder whether any of the mice with somatic *Vangl2* deletion (*Nkx1.2CreERT2*; *Vangl2^{F1/-}*; *Rosa26mTmG/+*) were allowed to be liveborn and presented with a spina bifida phenotype, and what was the prevalence of any spina bifida affected newborns. The lack of this information is a shortcoming of the manuscript.

We had originally decided not to collect litters at later stages of development because these are not mechanistically informative and the pre-spina bifida lesions we had seen convincingly demonstrated failed neural tube closure. While animal ethics regulations (UK Home Office) rightly prevent us causing pups to be born with spina bifida, we have now included images of later stage foetuses to demonstrate progression to open spina bifida (Figure 1K, shown below). We also confirmed that *Nkx1.2^{CreERT2/+} Vangl2^{F1/F1}* foetuses have dorsally kinked tails, a phenotype which is commonly associated with delayed PNP closure.

Figure 1: Mosaic *Vangl2* deletion causes spina bifida.

A,B. Representative PNP 3D reconstructions illustrating $Nkx1.2^{CreERT2}$ lineage tracing using the mTmG reporter. In **B**, non-recombined cells which continue to express tdTom are not shown. Scale = 100 μm , * indicates the zippering point (most caudal closed portion of NT).

C. Bright field image of an E9.5 mouse embryo indicating the location of the zippering point (*) and PNP (cyan arrowhead).

D. Surface-subtracted wholemount immunolocalisation of Vangl2 in the apical neuroepithelium of control (henceforth labelled “No Cre”) and Cre;Fl/- embryo, showing mosaic deletion (example regions demarcated by white dotted lines) in the latter. Scale = 20 μm . Whole-PNP images are 3D reconstructions, scale bar = 100 μm .

E. Quantification of Vangl2 immunolocalisation on cell borders lineage traced with EGFP or tdTom (not recombined) in Cre;Fl/Fl and Cre;Fl/- embryos. The average Vangl2 intensity in non-recombined (tdTom) cells was set at 100% in each embryo (4 embryos from independent litters per genotype). P values from ANOVA with post-hoc Bonferroni.

F. Optical cross-section through the neuroepithelium of a Cre;Fl/- embryo showing endogenous EGFP and immunofluorescently-detected Vangl2. White circles indicate borders between two EGFP+ cells, which are devoid of Vangl2 signal. Scale = 20 μm .

G. Quantification of the proportion of recombined neuroepithelial cells in E10.5 Cre;Fl/- embryos which closed or failed to close their PNP. Embryos with failed PNP closure have a median 16% of mutant neuroepithelial cells. Vangl2+ indicates the proportion of recombined cells in $Nkx1.2^{CreERT2/+}$ Vangl2^{+/+} Rosa26^{mTmG/+} embryos with wild-type Vangl2, which is not significantly different from either the Closed or Open groups with Vangl2-deleted cells. * $p < 0.05$ by Student's T-test.

H. Schematic representation of early (left) and late (right) PNPs illustrating progressive shortening and narrowing during physiological PNP closure. Cells in the caudal, open PNP (blue dots) translocate to the dorsal, closed PNP (blue arrows) during closure.

I. 3D reconstruction of the dorsal closed NT illustrating the region where EGFP+ neuroepithelial cells were analysed.

J. Representative images of two E10.5 Cre;Fl/- embryos with low (left) and high (right) recombination assessed from EGFP expression. Left: 37 somites, successful PNP closure has occurred. Right: 39 somites, tile-scanned image of failed PNP closure producing a pre-spina bifida lesion (white bracket). Dashed white boxes denote comparable regions of closed neural tube where recombination was quantified.

K. Representative images showing phenotypes in Cre;Fl/Fl and Cre;Fl/- embryos at E12.5. Cre;Fl/Fl embryos developed dorsally kinked tails (white arrow), whereas Cre;Fl/- embryos which fail to close their PNP also develop spina bifida (dashed black line in inset).

3. On Page 14, lines 8-9, the authors indicated that “Vangl2 deletion non-autonomously shortens microtubules tails in the mouse neuroepithelium placing Vangl2 upstream of microtubule organization.” It would be helpful if the authors provided additional details explaining just how the Vangl2 deletion non-autonomously regulated microtubule tails length puts Vangl2 upstream of microtubule organization.

We have now undertaken extensive additional experiments which identify the PCP effector ROCK as a regulator of neuroepithelial microtubule organisation. First, we observe that short-term ROCK inhibition increases apical microtubule density while reducing tail length in whole embryo culture. Next, we observed that apical ROCK intensity is diminished overall in the apical neuroepithelium of Cre;Fl/- embryos, but

Vangl2-deleted cells partially retain apical cap ROCK localisation. Thus, we propose that VANGL2/PCP signalling, which is known to act upstream of ROCK1 localisation, regulates microtubules in the neuroepithelium. This data is now in Figure 6, provided below.

Figure 6: ROCK activity enhances apical microtubules but shortens their apicobasal tails.

A-D) Wild-type embryos were cultured for 2 hours in vehicle or 10 μm of the ROCK inhibitor Y27632 (Y27). This time point was selected because we previously found it is sufficient to begin causing PNP widening¹⁰. **A)** Representative surface-subtracted images showing β -tubulin staining in vehicle and Y27-treated embryos. Microtubules appear more homogenous and denser in the Y27-treated embryos. **B)** Quantification of microtubule apical staining intensity showing

significantly higher intensity in Y27-treated embryos compared with vehicle controls. **C)** Optically-resliced cross-section showing microtubule apicobasal tails. **D)** Quantification of apicobasal tail length in vehicle and Y27-treated embryos, showing ROCK inhibition significantly shortens tail length. *P* values from Student's *T*-test.

E) Representative surface-subtracted images showing ROCK1 staining in the apical neuroepithelium of control and *Cre;Fl/-* littermate embryos. The *Cre;Fl/-* embryo image is shown with (right) and without (left) the *Vangl2*-deleted EGFP cells. Scale bar = 20 μ m.

F) Quantification of overall ROCK1 intensity in the apical neuroepithelium of control and *Cre;Fl/-* littermate embryos, demonstrating significantly lower signal in the latter. *P* value by Student's *T*-test.

G-H) Quantification of ROCK1 staining intensity selectively **G)** along cell borders or **H)** in the apical cap of neuroepithelial cells in control embryos lacking *Cre* or the indicated cell types in *Cre;Fl/-* embryos. These graphs are shown on the same scale to emphasize that ROCK1 border staining is much greater than apical cap staining, limiting visualisation of the latter. Points represent average values for cell types from individual embryos, lines link cell types from the same embryo. * *p* < 0.05 by mixed model analysis accounting for repeated measures from the same embryos.

I) Schematic illustration of ROCK's action. Refer to Figure 5 for colour coding. ROCK activity favours apicobasal microtubule tails over their apical networks. ROCK localises more intensely in the apical cap of *Vangl2*-deleted cells (green), which have diminished apical microtubules but longer tails than their neighbours.

All microscopy images were taken using AiryScan.

4. The underlying mechanism by which somatic *Vangl2* deletion regulates microtubule and cytoskeleton is not well studied. Single cell RNAseq analysis which can trace cell lineage and cluster *Vangl2*-replete and *Vangl2*-delete cells should be able to provide more information on the *Vangl2* somatic deletion effects on gene expression. While not integral to the manuscript, the authors might consider including scRNA seq in any revision.

We agree with the reviewer that scRNA Seq is not integral to this manuscript on non-autonomous regulation of apical constriction. We have now provided considerable additional mechanistic data, described above, identifying ROCK as a PCP effector which also regulates the microtubule cytoskeleton. scRNAseq will be considered for future work on this topic.

5. There appears to be an important omission in the author's introductory literature review. The first paper establishing the association between VANGL2 and human neural tube defects (N Engl J Med. 2010. PMID: 20558380) should be included in the references cited.

We apologise for this omission and have now added the reference.

--

Reviewer #2 (Remarks to the Author):

I suggest that the manuscript is acceptable for publication if minor comments below are addressed.

1. Figure 1 establishes the model that the authors use for mosaic Vangl2 deletion. It is unclear to me what is shown in panel C. I suggest that the authors add a schematic of the embryo and the structures that are referred to in panel C. Also a schematic showing the steps of NT closure across the developmental timepoints described in the manuscript would be useful.

Figure 1C has now been replaced by an overview image of an embryo indicating the location of the posterior neuropore. We have also included a schematic showing neuropore shortening and narrowing of characteristic of normal closure (Figure 1H).

2. Cre;Fl/- cellular orientation in Figure 2B is not randomised, instead there appears to be a preference for a RC direction in the Vangl2 Fl/-. Can the authors explain this effect?

While we appreciate the reviewer's observation that the proportion of Vangl2-deleted cells oriented in the 10° (RC) bracket appears high, the 60° bracket is equivalent. The overall distribution of orientations is not significantly different from the expected proportions in a random distribution. It would be tempting to speculate that cells may be preferentially rostrocaudally-oriented in the absence of VANGL2, but we cannot reach that conclusion with the data available.

3. In Figure 3, the authors use laser ablation to show that clusters of cells that consist of Vangl2 deficient cells and their GFP negative neighbours are under reduced apical tension and that this effect is due to non-cell autonomous changes in tension in the neighbours. It would be interesting if the authors could re-analyse the mixed clusters to examine if there is a direct correlation between the GFP+/GFP- ratio and retraction range upon ablation? If a cluster consists of only one GFP+ cell you would expect a greater loss in retraction compared to a cluster that is mostly GFP+?

Please refer to our answer above to Reviewer 1's second comment. In that response we describe new analyses which lead us to favour an "all-or-none" mode of non-autonomous suppression of apical constriction.

4. Is it also possible that neighbouring GPF negative cells are actively extruding the Vangl2 deficient cell which results in enhanced cap MHC-II and longer microtubule tails? This would likely also results in overall smaller Vangl2 deficient cells and larger neighbours, as shown in Figure 2?

We thank the reviewer for this exciting suggestion. If Vangl2-deleted cells were actively extruded, that may have led to a rescue strategy whereby enhancement of their extrusion might have prevented spina bifida in patients with somatic mutations in PCP genes. Active extrusion should reduce the number of Vangl2-deleted cells in the neuroepithelium. To test this hypothesis we generated *Nkx1.2^{CreERT/+} Vangl2^{+/+} Rosa26^{mTmG/+}* embryos in order to compare lineage-traced cell proportions with cells which are otherwise WT. We found no significant difference between the proportion of neuroepithelial cells lineage traced in *Nkx1.2^{CreERT/+} Vangl2^{+/+} Rosa26^{mTmG/+}* embryos with WT Vangl2 compared with *Nkx1.2^{CreERT/+} Vangl2^{Fl/-} Rosa26^{mTmG/+}* embryos lacking Vangl2, negating the hypothesis that Vangl2-deleted cells are extruded. This is now included in Figure 1G (please refer to Figure 1 shown above).

--

Reviewer #3 (Remarks to the Author):

However, the study does not identify a clear mechanistic link between Vangl2 and its downstream effects on the cytoskeleton and apical constriction. For example, how does Vangl2 affect myosin localization and microtubule organization? Does Vangl2 regulate myosin and microtubules via independent or common pathways?

We now provide additional mechanistic data which confirms VANGL2/PCP may indeed regulate the actomyosin and microtubule cytoskeletons through a single pathway involving ROCK. We and others had previously demonstrated that ROCK activity is required for neuroepithelial apical constriction (e.g. Butler and Short et al, J Cell Sci, 2019). Here we observe that apical ROCK intensity is diminished overall in the apical neuroepithelium of Cre;Fl/- embryos, but Vangl2-deleted cells partially retain apical cap ROCK localisation. We also observe that short-term ROCK inhibition increases apical microtubule density while reducing tail length in whole embryo culture. Thus, we can propose that VANGL2/PCP signalling acts upstream of ROCK, which regulates microtubule organisation (please see Figure 6 provided above).

Secondly, it remains unclear how Vangl2-deleted cells can affect apical constriction of neighboring cells, while retaining the ability to constrict themselves. How does PCP activity change in the cells adjacent to Vangl2-deleted cells? If PCP signaling is altered in both types of cells, what regulates the differing apical constriction outcomes in these cells?

Perhaps the authors could check previous work in other in vivo systems showing that disruption of apical constriction in one cell affects the ability of neighbouring cells to constrict (see for instance Samarage et al Dev Cell).

We report three mechanisms by which *Vangl2*-deleted cells may constrict more than their neighbours:

- 1) Having less dense apical microtubules is expected to reduce the resistance to constriction *Vangl2*-deleted cells must overcome.
- 2) Our new analyses of ROCK1 staining shows *Vangl2*-deleted cells have higher apical cap staining than their neighbours, or indeed than distant cells. We hope to study the regulation and function of ROCK and MHC-IIb in different cell sub-domains in future work.
- 3) Being primarily surrounded by non-constricting neighbours may enable *Vangl2*-deleted cells to constrict with less resistance.

We also thank the reviewer for their question about whether constricting *Vangl2*-deleted cells interfere with their neighbours' constriction, and the related question below about whether the change in apical area of *Vangl2*-deleted cells correlates with that of their neighbours. To address this we reanalysed our live-imaging data, finding that the change in area of most *Vangl2*-deleted:Neighbour cell pairs is only weakly correlated and that both positive and negative correlations are observed. We therefore conclude that the constriction of *Vangl2*-deleted cells does not predictably cause their replete neighbours to dilate. This is now shown in Supplementary Figure 8 (below).

Supplementary Figure 8: Changes in apical areas of live-imaged *Vangl2*-deleted cells are not correlated with that of their *Vangl2*-replete neighbours.

A. Live-imaged apical surface of a *Cre;Fl/-* embryo showing constriction of the *Vangl2*-deleted cell (green) without substantial change in the apical area of its *Vangl2*-replete neighbours (grey outlines, illustrated in maroon in the bottom panel). Panels ~5 minutes apart, scale bar = 10 μ m.

B. Quantification of apical area in a representative *Vangl2*-deleted (EGFP) and neighbouring *Vangl2*-replete pair over time.

C. Correlation coefficient of changes in apical areas over time (as in B) from 30 EGFP-Neighbour cell pairs from 3 independent embryos, showing generally weak correlation.

Other comments:

- Although Vangl2 is indeed a core component of the PCP signaling pathway, can the authors better demonstrate whether the aberrant phenotypes observed in the mosaic Vangl2-deleted neuroepithelium directly result from impaired PCP signaling, or if they stem from other PCP-independent functions of Vangl2?

Our new findings implicating the known PCP effector ROCK (Figure 6) are consistent with VANGL2 acting through its conventional PCP roles in this context.

- Do the Vangl2-deleted cell neighbours undergo a corresponding expansion of their apical surfaces when the Vangl2-deleted cells constrict?

Please refer to the new data (Supplementary Figure 8) provided in response to the reviewer's second point, above, showing that neighbour cells do not predictably dilate as the Vangl2-deleted cells constrict.

- The analysis of the three types of cell junctions (EGFP/neighbour, neighbour/neighbour, and distant/distant) for MHC-IIb levels could be supplemented with laser ablation experiments to confirm the differences in junctional tension.

We previously explained (Butler and Short et al, J Cell Sci, 2019) that ablation of individual apical cell borders is not currently feasible. Nevertheless, we have now undertaken new analyses quantifying the apical area of a large number of Vangl2-deleted, neighbouring and distant cells as a surrogate of constriction. We confirm that Vangl2-deleted cells have smaller (more constricted) apical surfaces than their neighbours. Please refer to the data provided in Supplementary Figure 5 shown in response to Reviewer 1's second comment.

- In Vangl2-deleted cells, MHC-IIb is primarily cortical (instead of being enriched at the apical cap). Do the total MHC-IIb levels differ between Vangl2-deleted and control cells, or are the levels unchanged and only the subcellular distribution of MHC-IIb is altered?

We have now provided Supplementary Figure 9, which shows that the overall staining intensity of MHC-IIb in the apical neuroepithelium is not significantly different between Cre;Fl/- embryos and their littermates lacking Cre. However, apical cap staining within each neuroepithelium shows less cell-to-cell variability in Cre;Fl/- embryos, further supporting a role for VANGL2 signalling in myosin localisation. This is shown below:

Supplementary Figure 9: Mosaic Vangl2 deletion does not significantly alter apical neuroepithelial MHC-IIb intensity, but reduces its variability between cells on the apical cap.

A. Maximum-projected images of a control and littermate Cre;Fl/- embryo, each at the 23 somite stage, wholemount stained to show MHC-IIb. Arrows indicate the neural fold F-actin cables, scale bar = 100 μ m.

B. Quantification of overall MHC-IIb staining intensity, which is not significantly different between control and Cre;Fl/- embryos. Points represent individual embryos.

C. The coefficient of variation (C.V.) of MHC-IIb apical cap intensity was quantified in control and Cre;Fl/- embryos as a measure of variability in apical cap myosin intensity. This variability results from cells having low (#) or high () apical cap staining, as shown in the inset.*

--

Reviewer #4 (Remarks to the Author):

The main shortcomings are slightly inadequate description and discussion of their genetic system, which is very complex.

We have now included text in the Methods which provides a more detailed description of the genetic tools used:

“Four transgenic alleles were used in this manuscript. *Rosa26^{mTmG}* (mTmG, Ref) is a reporter allele which causes all cells to express membrane tdTomato red fluorescent protein, but is recombined by Cre to express membrane EGFP instead. The conditional *Vangl2* floxed (*Vangl2^{Fl}*) allele was as previously described and a *Vangl2*-null allele was generated by crossing to *Actin^{Cre}* and subsequent breeding out of that ubiquitous Cre more than ten generations before the start of these studies. Two conditional Cre drivers were used in this study. *Nkx1-2^{CreERT2}* is a tamoxifen-inducible Cre driver which recombines in axial progenitor cells, persistently lineage-tracing the neuroepithelium. Stud males with *Nkx1-2^{CreERT2/+}* and *Vangl2^{Fl/-}* alleles were bred with females harbouring conditional *Vangl2* alleles as well as homozygous mTmG reporters. This cross was used to lineage-trace *Vangl2*-deleted cells at least 24 hours after tamoxifen administration. The second conditional Cre driver used was *Grhl3^{Cre}* as previously described. This Cre recombines throughout the surface ectoderm, and in an increasing proportion of neuroepithelial cells. Activation of neuroepithelial *Grhl3^{Cre}* is not temporally restricted, but predictably produces mosaic recombination at later stages of posterior neuropore closure. All mouse colonies were maintained in house on a C57Bl/6 background and bred from 8 weeks of age. Cre-negative littermates generated in the course of colony maintenance were used as WT controls for embryo culture.”

The description of “sarcomere-like” assembly of myosin at the apical surface of cells has little merit here and should be omitted, opting instead for language that implies less about the very-unclear structure of actomyosin at that domain.

The term “sarcomere-like” is no longer used to describe MHC-IIb staining patterns observed in this manuscript.

Specific comments:

1. Page 3, lines 15-20: The citations here are useful and interesting but come from a variety of animals and tissues. I’d appreciate clarity on the systems these phenomena occur in as none of them are neural tube closure and that isn’t made clear.

This section has been amended to specify the species or tissues described.

2. Page 4, lines 26-27: Is it known or is there a hypothesis why *Grhl3*-Cre drives patchy

recombination in the neural epithelium? Is Cre expressed at low levels across the neural epithelium or is the expression of Cre itself patchy?

We have added the text below to describe the expression pattern of the endogenous *Grhl3* gene and the knock-in *Grhl3^{Cre}* allele. We have extensive experience using this Cre, which produces mosaic neuroepithelial recombination using a variety of reporters (mTmG, YFP, LacZ). The text below has been added to the results section:

“Neuroepithelial expression of the *Grhl3* gene is very low and patchy at E8.5, when it is largely restricted to the surface ectoderm overlying the neural folds, but increases at later stages². Consistent with this, the knock-in *Grhl3^{Cre}* lineage-traces a small proportion of neuroepithelial cells as early as the 8-somite stage and this proportion increases during development^{1,3,4}. *Grhl3^{Cre/+}Vangl2^{Fl/Fl}* embryos show patchy loss of neuroepithelial VANGL2 protein at later stages of development, beyond the ~19-somite stage, but not earlier (Supplementary Figure 4A-C).”

3. Figure 1A/B: would appreciate if the anteroposterior axis was labeled.

Annotation of anatomical direction has now been added to Figure 1.

Figure 1G: It is unclear how EGFP expression was compared between the two embryos. Given the one on the left closed, was EGFP expression in the neural epithelium inferred from the surface epithelium? Do you have evidence that surface and neural recombination levels correlate?

We used confocal microscopy to image below the surface ectoderm to the underlying neuroepithelium of the closed neural tube. EGFP recombination was compared between embryos with a fully closed neural tube and littermates which had failed to fully close. Cells in the ventral posterior neuropore are known to translocate to this region during closure, as is now schematically illustrated in Figure 1H. A 3D reconstruction showing EGFP-labelled neuroepithelial cells in the dorsal, closed neural tube has also been added in Figure 1I (please refer to this image shown above).

4. Figure 2: I would like to see a comparison of the shape of the Cre;Fl/-, EGFP+ to the other conditions, perhaps via aspect ratio. Also, “EGF” should be replaced by “EGFP”

The EGFP title has been corrected.

The cell apical area data provided in Figure 2 was based on data generated before we understood the non-autonomous effects *Vangl2*-deleted cells have. We have now independently replicated that analysis in order to separate out neighbouring and distant cells. This confirmed that *Vangl2*-deleted cells have a smaller apical area than their neighbours, but that apical aspect ratio, circularity and shape index are not significantly

different between the cell types analysed. This new data is shown in Supplementary Figure 5, which is shown above in response to Reviewer 1's second comment.

5. Figure 3: Would like to see EGFP+/- arranged in the same order as previous figures

Figure 3C has been changed so that data from clusters with EGFP+ cells are shown before those without EGFP.

6. Figure 4: 4B is confusing but I think I get it...

This figure provides a visual representation showing that one group - the neighbouring cells - appears distinct from the others. We have amended the axis labels in this figure to make it easier to interpret.

7. Page 7/Methods/Supplementary Figure 5: Would like to see mention somewhere of what imaging marker was used in the live-imaging experiments.

This has now been added to the Methods:

“Endogenous membrane-localised tdTomato or EGFP expressed from the mTmG locus was visualised.”

8. Page 9, lines 6-19: the myosin arrangement could possibly be “sarcomere-like” but I think it is premature to describe it as such based on the presented data. The cited results from Drosophila showed a) a fairly different localization pattern with radialized cell polarity at the apical surface and b) more data to demonstrate a “sarcomere-like” actomyosin structure at the apical surface.

The term “sarcomere-like” is no longer used to describe MHC-IIb staining patterns observed in this manuscript.

9. Page 11, lines 11-25: This is useful information but I wonder if it wouldn't belong better at the beginning of the results section, as it addresses technical information that affects the results and generated the concept. In particular, I think the point that Grhl3-Cre induces Vangl2 recombination relatively late in neural tube closure is a point that should be made clearer early on.

We have now extended our analysis of the *Grhl3*^{Cre} model, alongside our main studies using *Nkx1.2*^{CreERT2}. We provide new data showing late VANGL2 reduction and diminished apical constriction in *Grhl3*^{Cre/+} *Vangl2*^{F1/F1} embryos in Supplementary Figure 4 (please refer to the figure shown above).

References:

- 1 Galea, G. L. *et al. Dis Model Mech* **11**, doi:10.1242/dmm.032219 (2018).
- 2 De Castro, S. C. P. *et al. Dev Biol* **435**, 130-137, doi:10.1016/j.ydbio.2018.01.016 (2018).
- 3 Mole, M. A. *et al. Dev Cell* **52**, 321-334 e326, doi:10.1016/j.devcel.2020.01.012 (2020).
- 4 Rolo, A. *et al. Elife* **5**, e13273, doi:10.7554/eLife.13273 (2016).
- 5 Galea, G. L. *et al. Proc Natl Acad Sci U S A* **114**, E5177-E5186, doi:10.1073/pnas.1700934114 (2017).
- 6 Juriloff, D. M. & Harris, M. J. *Birth Defects Res A Clin Mol Teratol* **94**, 824-840, doi:10.1002/bdra.23079 (2012).
- 7 Wang, L. *et al. Mol Genet Metab* **124**, 94-100, doi:10.1016/j.ymgme.2018.03.005 (2018).
- 8 Humphries, A. C. *et al. Elife* **9**, doi:10.7554/eLife.53532 (2020).
- 9 Tian, T. *et al. Hum Genet*, doi:10.1007/s00439-020-02172-0 (2020).
- 10 Butler, M. B. *et al. J Cell Sci* **132**, doi:10.1242/jcs.230300 (2019).

Reviewers' Comments:

Reviewer #1:

Remarks to the Author:

The revised manuscript adequately address all of the concerns I raised in my initial review. The authors performed additional experiments, some I requested and others that were rational follow up studies and the manuscript now is a wonderful contribution to the neural tube literature.

Reviewer #2:

Remarks to the Author:

The revised manuscript by de Galea et al has addressed my concerns.

I appreciate that the authors have performed extensive additional experiments of great quality that have really improved the story. I especially think the examination of the role of ROCK in microtubule organization which in a new main Figure is a great addition.

Reviewer #3:

Remarks to the Author:

The revised manuscript has been improved and the authors have satisfactorily addressed most of my initial comments. But there are three remaining points that need to be addressed or clarified:

- 1) The authors now provide additional experiments showing that ROCK inhibition alters apical microtubule density / tail lengths. It would be important to perform some control experiments demonstrating that the Y27632 inhibitor is indeed affecting ROCK activity, as this drug has several other known targets. Alternatively, they could have tried H1152 as independent approach.
- 2) They also claim that apical ROCK intensity is generally reduced in the neuroepithelium of Cre;Fl-embryos, but EGFP+ cells still retain higher apical ROCK levels over their neighbours. However, these differences are not immediately evident from the images provided in Fig 6E, especially the comparison between EGFP+ cells and neighbours in the Cre;Fl- epithelium. Can the authors provide zoomed views of the EGFP+ cells and their neighbours for a more clear comparison of the apical ROCK levels?
- 3) The graph in Fig 6H shows significant overlap/spread in the intensity values obtained for all the groups (No Cre, EGFP, Nei., Dist.). Is there a significant difference in apical ROCK intensity between the No Cre and Cre epithelium (please provide a p-value in the graph)?

Reviewer #4:

Remarks to the Author:

My concerns have been addressed and i now support publication.

REVIEWERS' COMMENTS

Reviewer #1 (Remarks to the Author):

The revised manuscript adequately address all of the concerns I raised in my initial review. The authors performed additional experiments, some I requested and others that were rational follow up studies and the manuscript now is a wonderful contribution to the neural tube literature.

No response required.

Reviewer #2 (Remarks to the Author):

The revised manuscript by de Galea et al has addressed my concerns.

I appreciate that the authors have performed extensive additional experiments of great quality that have really improved the story. I especially think the examination of the role of ROCK in microtubule organization which in a new main Figure is a great addition.

No response required.

Reviewer #3 (Remarks to the Author):

The revised manuscript has been improved and the authors have satisfactorily addressed most of my initial comments. But there are three remaining points that need to be addressed or clarified:

1) The authors now provide additional experiments showing that ROCK inhibition alters apical microtubule density / tail lengths. It would be important to perform some control experiments demonstrating that the Y27632 inhibitor is indeed affecting ROCK activity, as this drug has several other known targets. Alternatively, they could have tried H1152 as independent approach.

We have already performed and reported control experiments for the Y27632 inhibitor requested by the reviewer. These have now been described in the text of the current manuscript:

Page 11, line 25: We previously reported that the commonly-used compound Y27632 diminishes Rho/ROCK signalling by reducing active Rho in mouse whole embryo culture, preventing down-stream responses including restricted apical localisation of actomyosin and apical constriction of the neuroepithelium^{22,41}. Our previously-published

time course experiments show that Y27632 treatment begins to cause PNP widening within two hours of treatment in mouse whole embryo culture²².

2) They also claim that apical ROCK intensity is generally reduced in the neuroepithelium of Cre;Fl- embryos, but EGFP+ cells still retain higher apical ROCK levels over their neighbours. However, these differences are not immediately evident from the images provided in Fig 6E, especially the comparison between EGFP+ cells and neighbours in the Cre;Fl- epithelium. Can the authors provide zoomed views of the EGFP+ cells and their neighbours for a more clear comparison of the apical ROCK levels?

A zoomed in image has now been provided (Figure 6i). The reason the apical enrichment of ROCK is not obvious is due to its levels being of lower intensity than the cell cortex localization, as was shown by the axes of graphs in Figure 6g and h. To circumvent this, we have shown Figure 6i with a fire LUT to emphasize both low and high intensity localisations.

3) The graph in Fig 6H shows significant overlap/spread in the intensity values obtained for all the groups (No Cre, EGFP, Nei., Dist.). Is there a significant difference in apical ROCK intensity between the No Cre and Cre epithelium (please provide a p-value in the graph)?

A p value has been added to indicate that the only significant differences in this graph are between cell types in Cre-positive embryos, not between Cre and No Cre.

Reviewer #4 (Remarks to the Author):

My concerns have been addressed and i now support publication.

No response required.